# WinQ: Accelerating Quantization-Aware Training of Language Models Around Saddle Points

Dongyue Li [1]  Zechun Liu [2]  Kai Yi [2]  Zhenshuo Zhang [1]  Changsheng Zhao [2]  Raghuraman Krishnamoorthi [2]
Harshit Khaitan [2]  Hongyang R. Zhang [1]  Steven Li [2]

## Abstract

Quantization-aware training (QAT) is widely adopted to quantize language models by training full-precision weights using gradients from the quantized model. The main bottleneck is its slow convergence and early performance plateau, particularly below 4-bit-widths. While this problem has been observed in prior work, its precise cause remains unclear. In this paper, we analyze the convergence of QAT by estimating the spectrum of the loss-surface Hessians. We find that the weights converge to flat regions around saddle points, where a large fraction of the Hessian eigenvalues are both positive and negative. During training, an increasing fraction of Hessian eigenvalues concentrates around zero, whose magnitude decreases. At lower bit-widths, the magnitude of eigenvalues in the Hessian spectrum is significantly smaller. To mitigate these issues, we propose an algorithm called WINQ to accelerate QAT, which involves: (1) periodically resetting weights to the linear interpolation of full-precision and quantized weights, reducing the distance to the quantization grid and increasing eigenvalue magnitude, and (2) computing gradients of noise-injected weights to regularize the Hessian. Extensive experiments show that WINQ accelerates QAT by up to $4\times$ across various quantization methods and models. Under the same training cost, WINQ improves state-of-the-art sub-4-bit quantization by up to 8.8%. These results are consistent across 16 settings with different language models, quantization methods, and bit widths.

[1]Northeastern University, MA [2]Meta AI, CA. Correspondence to: Dongyue Li <li.dongyu@northeastern.edu>, Hongyang Zhang <ho.zhang@northeastern.edu>, Steven Li <stevenlx@meta.com>.

*Proceedings of the 43rd International Conference on Machine Learning*, Seoul, South Korea. PMLR 306, 2026. Copyright 2026 by the author(s).

## 1. Introduction

Quantization is a key technique for efficiently deploying large language models by representing weights and activations with lower precision. Since directly converting a trained model to lower precision often increases loss, quantization-aware training (QAT) (Jacob et al., 2018) is introduced to mitigate quantization-induced loss. This approach works by training the full-precision model weights while simultaneously applying quantization to the weights and, in some cases, to activations. This training approach has significantly advanced the language model performance in extremely low precision. Recent developments in quantization-aware training have enabled language models to operate at sub-4-bit precision with performance close to full precision (Gholami et al., 2022), whereas post-training quantization methods remain significantly less accurate below 4-bit precision (Ashkboos et al., 2024).

A major bottleneck of quantization-aware training is slow convergence and early plateauing of the test loss, which leads to substantial computational cost and limits further improvement. For example, starting from a pretrained model, the cost of 4-bit quantized training remains around 10% of full-precision pretraining (Liu et al., 2025b). Training with lower-bit-width quantization, such as 1-bit, is even more difficult, as convergence is remarkably slower (Wang et al., 2023). It typically requires training on at least 10B tokens to achieve 1-bit precision, even for language models with 100M parameters (Panferov et al., 2025). Further increasing the training cost yields limited gains according to the scaling law (Kumar et al., 2025). This work aims to understand the reasons behind slow convergence and to propose methods to accelerate quantized training for language models. Our objective is to reduce the cost required to reach a given training loss while achieving a better test loss under the same budget.

Prior works have focused on improving the design of quantization methods and gradient estimation in quantized training, while the problem of slow convergence has not been studied in depth. Uniform quantization is the most widely used quantization method. Due to its non-differentiability, a technique named straight-through estimation (STE) (Ben-

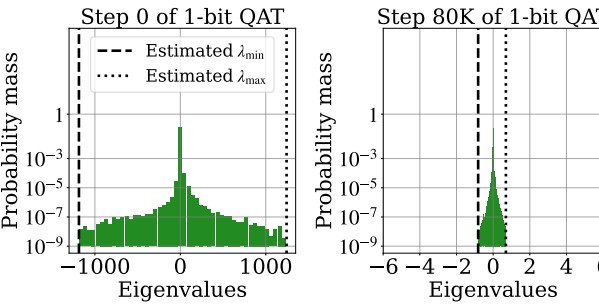
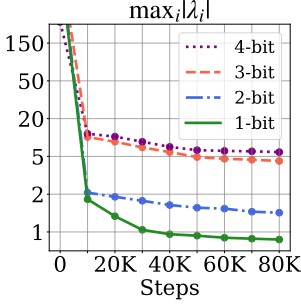
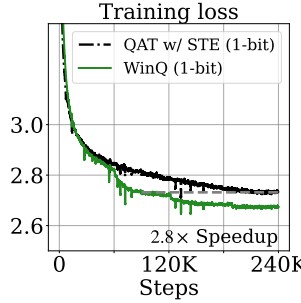

*(a)* The estimated empirical eigenvalue distribution of the loss Hessian at step 0 and 80K in 1-bit QAT

*(b)* Maximum magnitude of Hessian eigenvalues

*(c)* Comparing our method with the state-of-the-art

*Figure 1.* We investigate the slow convergence of quantization-aware training by analyzing the loss Hessian with respect to model weights. Figure 1a: By estimating the Hessian spectrum, we find that slow convergence is mainly because the model weights converge to a flat surface around saddle points, where the rate is governed by the magnitude of Hessian eigenvalues. In low precision, over 40% of the Hessian eigenvalues are near zero, and the maximum absolute eigenvalue decreases markedly during training. Figure 1b: We further observe that at lower bit-widths, Hessian eigenvalues have much smaller magnitudes, consistent with the slower convergence in these settings. Figure 1c: We propose WINQ, which integrates a weight interpolation technique with noise injection. Our approach applies to various quantized training methods with minimal overhead and significantly speeds up SoTA methods at extremely low precision.

gio et al., 2013) is often used to estimate gradients by copying the gradients of quantized weights to the full-precision weights. Building on this prior practice, a recent work, ParetoQ (Liu et al., 2025b), reduces quantization error in weights with a new stretched elastic quantization method and learnable step sizes (Esser et al., 2020). QuEST (Panferov et al., 2025) improves the quantization error with a normalizing Hadamard Transform and refines gradient estimation by masking the gradients of weights with high quantization errors. Yet, improving the slow convergence of training has been left open in prior works.

In this paper, we examine the slow convergence of quantization-aware training by analyzing the loss Hessian of the quantized model. Our motivating observation is that the gradient norm of quantized training often drops near zero. In this region, the convergence rate is determined by the local curvature of the loss surface. Thus, we hypothesize that at low precision, the loss Hessian matrix exhibits many small—or even zero—eigenvalues, thereby slowing convergence. In contrast to prior work on using second-order information to guide the design of quantization methods (Dong et al., 2019; 2020), our work further examines the full spectrum of the Hessian of the loss surface.

To this end, we first estimate the Hessian spectrum of quantized language models using Hessian–vector products (Bai et al., 1996; Ghorbani et al., 2019). Our main finding is that across 1- to 4-bit training, model weights converge to a flat region near saddle points. As illustrated in Figure 1a, the Hessian exhibits a balanced mix of positive and negative eigenvalues, with over 40% concentrated near zero. Provided that the gradient norm is approaching zero, this indicates that the weights are stuck around saddle points, where the convergence speed is governed by the maxi-

mum magnitude of Hessian eigenvalues. After the initial training phase, the magnitude of the eigenvalues decreases markedly, leading to slow convergence. Further, as shown in Figure 1b, the magnitude of eigenvalues becomes even smaller at lower bit-widths, consistent with the slower convergence observed in lower-bit precision. Our findings build on a line of recent work that establishes a connection between the Hessian spectrum and learning in neural networks (Ju et al., 2022; 2023; Zhang et al., 2024a; 2026a), while offering a new perspective on the slow convergence of quantized training and the saddle-point problem in high-dimensional non-convex optimization.

In this work, we design an approach for QAT with negligible additional cost. Our approach is based on a weight re-initialization technique where model weights are periodically reset to a linear interpolation between $W$ and $Q(W)$, denoted as $(1 - \alpha)W + \alpha Q(W)$ for an $\alpha$ between 0 and 1. This design provably reduces the distance between the full-precision and quantized weights by acting as a proximal update step on an $\ell_2$-regularized training objective. We show that this step increases the magnitude of Hessian eigenvalues, thereby accelerating convergence. Further, we incorporate a noise injection method that computes the gradient after perturbing $W$ with a random Gaussian noise vector $U$ at each iteration, inspired by prior work (Jin et al., 2017) to accelerate training around saddle points. Taken together, our approach, named WINQ, can be applied on top of various quantization methods, including those using the Hadamard Transform.

We extensively evaluate WINQ across multiple bit-widths and language models. First, at precisions below 4-bit, our method achieves **1.5–4×** speedups over state-of-the-art quantized training methods, as shown in Figure 1c. Sec-

ond, at the same computational cost, our approach improves sub-4-bit quantization performance by up to **8.8**%. Across 16 settings, we observe consistent gains for LLaMA and Qwen models with 0.6B to 3B parameters. Third, WINQ can also be applied to quantization methods using the Hadamard Transform, further boosting performance by up to **2.8**%. Finally, ablation studies confirm that both components in our algorithm are essential to the performance. The code for reproducing this work is available at https://github.com/facebookresearch/WinQ.

In summary, this paper makes three contributions to quantized training for low-precision LLMs:

- First, we analyze the slow convergence of quantized training through the loss Hessian and identify that the primary reason arises from model weights converging to a flat surface near saddle points.

- Second, we propose a generic approach to accelerate quantized training with negligible computational overhead by combining a novel weight re-initialization technique and noise injection.

- Third, we demonstrate that our approach significantly accelerates convergence and improves test performance for language models across a range of sub-4-bit quantization methods.

## 2. Preliminaries

We describe a formulation for quantization-aware training. Let $f_W$ denote a language model with parameters $W \in \mathbb{R}^d$. Let $\hat{L}_W$ denote its empirical risk on a training set. Let $Q(\cdot)$ be a quantization function that maps the model parameters to weights that can be represented by lower precisions. We denote its output weights as $Q(W)$. Quantization-aware training minimizes the loss $\hat{L}_{Q(W)}$ with respect to $W$ through computing $Q(W)$ at each iteration. Thus, $W$ is also called latent weights.

As $Q(\cdot)$ is typically non-differentiable, quantized training algorithms often use the *straight-through estimator* (Bengio et al., 2013), which copies the gradient of $Q(W)$ as the gradient of $W$. After training, the final latent weights $\hat{W}$ are quantized to $Q(\hat{W})$ for inference. We provide precise definitions of other terminologies, such as saddle points, in Appendix A. Next, we describe the widely used uniform quantization, which can be instantiated in many ways.

**Definition 2.1** (Uniform quantization). Given a bit-width $n$, uniform quantization linearly scales the weights and rounds them to the nearest integers:

$$Q(W) = a \left\lfloor \text{clip}\left(\frac{W - b}{a}, v_{\text{neg}}, v_{\text{pos}}\right) \right\rceil + b, \quad (1)$$

where $a$ is the scale, $b$ is the bias, and $\lfloor \rceil$ is a rounding function that converts the value to its nearest integer.

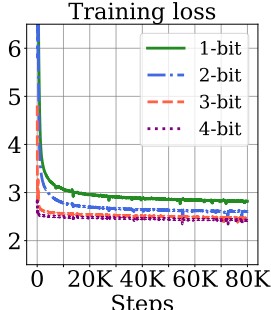
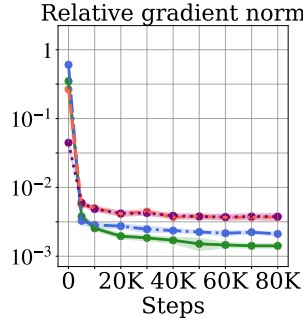

*Figure 2.* The convergence of the training loss and the relative gradient norm in quantization-aware training from 1-bit to 4-bit precision.

clip$(W, v_{\text{neg}}, v_{\text{pos}})$ clamps the values in $W$ to a range between $v_{\text{neg}}$ and $v_{\text{pos}}$, where the bounds are determined by the quantization bits.

For more concrete examples, in symmetric min-max quantization, $a$ is set by the maximum absolute weight: $a = \frac{\max_i |W_i|}{2^{n-1} - 1}$, with $v_{\text{neg}} = -2^{n-1}$, $v_{\text{pos}} = 2^{n-1} - 1$, and $b = 0$. In asymmetric min-max quantization, the scale is based on the full range of weights: $a = \frac{\max_i W_i - \min_i W_i}{2^n - 1}$, with $v_{\text{neg}} = 0$, $v_{\text{pos}} = 2^n - 1$, and $b = \min_i W_i$. In addition, the scale $a$ can also be treated as a learnable parameter and optimized jointly with the model (Esser et al., 2020).

**Convergence rate under varied bit-width.** We present empirical observations of the convergence speed using a state-of-the-art QAT method based on STE (Liu et al., 2025b). We perform quantization-aware training for the pretrained LLaMA-3-1B on the FineWebEdu dataset, using AdamW as the optimizer.

In Figure 2, we first illustrate the training convergence in 1-bit, 2-bit, 3-bit, and 4-bit, respectively. We find that the convergence slows down after the initial 10K training steps. QAT in 2-bit and 1-bit converges significantly slower than 4-bit quantization. Further, we evaluate the gradient norm relative to the weight norm, i.e., $\|\nabla_W \hat{L}_{Q(W)}\|_2 / \|W\|_2$, along training, which converges to the scale between $10^{-3}$ and $10^{-2}$ across 1-bit to 4-bit precision.

Near a first-order stationary point, the convergence rate of gradient-based methods is governed by the magnitude of Hessian eigenvalues. One hypothesis is that low-bit quantization tends to yield flat curvature in the weight space. Additionally, we observe that the distance between $W$ and $Q(W)$ increases with lower precision, suggesting greater difficulty in training at lower precision. The relative norm of quantization error, i.e., $\|Q(W) - W\|_2 / \|W\|_2$, increases from 16% in 4-bit to 70% in 1-bit. This motivates us to study the connection between the slow convergence of quantized training and flat curvature in the loss surface.

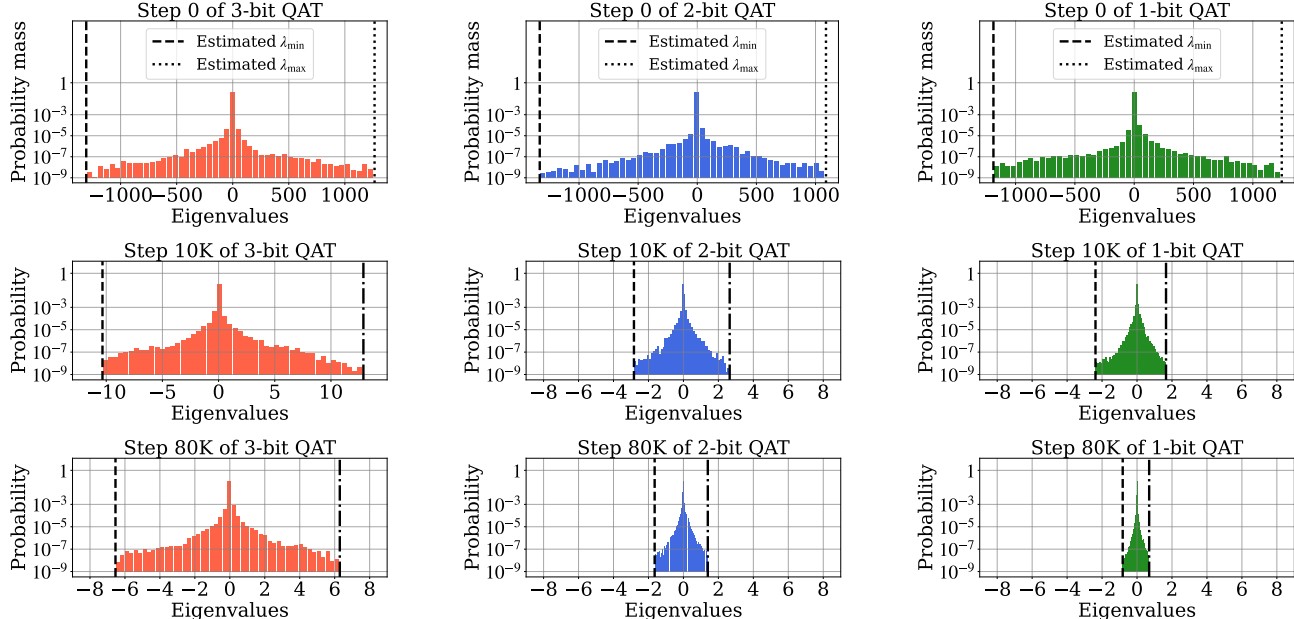

*Figure 3.* We illustrate the estimated eigenvalues and their probability mass of the loss Hessian matrix in 3-bit, 2-bit, and 1-bit QAT, respectively. (1) Most of the eigenvalues concentrate close to zero, and there are both negative and positive eigenvalues with comparable probability mass. This suggests that the model weights converge to a flat region around saddle points during training. (2) For a lower precision, the magnitude of eigenvalues becomes smaller, and more eigenvalues are around zero. This suggests that the model weights are in a region with flatter curvature, thus exhibiting slower convergence. The $x$-**axis** corresponds to the eigenvalues. $y$-**axis** corresponds to the probability mass function of the eigenvalues (in log scale). We illustrate the eigenvalues for the weights in transformer layers. We describe the results for the FP16-precision model and for embedding layers in Appendix B.

## 3. Our Approach

In this section, we analyze the Hessian spectrum of the loss surface during quantized training. We find that the Hessian exhibits a concentration of zero eigenvalues, with small magnitudes for both negative and positive eigenvalues. Thus, the slow convergence can be attributed to the presence of saddle points, which are more pronounced at lower precision. To tackle this, we find that a weight reinitialization technique significantly speeds up training by linearly interpolating between full-precision and quantized weights. We propose an approach that further incorporates noise injection into the model weights during training, thereby improving and speeding up various quantization methods.

### 3.1. Measuring Hessian Spectrum

We first present our analysis of the loss Hessian with regard to model weights. Specifically, we estimate the Hessian spectrum, i.e., the eigenvalue and its probability mass over all eigenvalues. As fully computing the Hessian matrix is infeasible, we use a numerical method called Stochastic Lanczos Quadrature (Bai et al., 1996; Chen et al., 2021), which can be implemented using Hessian-vector products. We present results using LLaMA-3-1B trained with 1-bit to 4-bit precisions and estimate the loss Hessian on the train-

ing data. See implementation details in Appendix A.

**Saddle points in quantization-aware training.** We identify that the model weights get stuck in a flat region near saddle points. Figure 3 illustrates the estimated eigenvalue distributions in 1-bit, 2-bit, and 3-bit weight precision, respectively. Consistently across three precisions, we observe both negative and positive eigenvalues in the Hessian matrix, with comparable probability mass. As training progresses, more eigenvalues concentrate near zero, and the magnitude of eigenvalues decreases significantly. For example, in 3-bit QAT, the probability mass of zero eigenvalues increases from **7**% at step 0 to **41**% at 80K steps. Numerically, we regard the estimated eigenvalues in a range between $-10^{-3}$ and $10^{-3}$ as zero. See Appendix A for a precise definition of saddle points.

**Flatter curvature in lower bit-width.** We find that the model weights are stuck in a region with flatter curvature for lower precision, resulting in slower convergence of quantization-aware training. In Figure 3, comparing among the three bit-widths, we found that the magnitude of the eigenvalues becomes much smaller, from **6** in 3-bit to **2** in 2-bit and less than **1** in 1-bit at 80K steps. Additionally, a larger proportion of eigenvalues is near zero. The probability mass for zero eigenvalues increases from **41**% in 3-bit to **55**% in 2-bit and **63**% in 1-bit.

**Effect of activation quantization.** We further evaluate the Hessian spectrum under 8-bit and 4-bit activation quantization using Llama-1B with 1-bit weight quantization. We find that training with lower-precision activations (8-bit and 4-bit) produces (**23%** and **32%**) more smaller-magnitude Hessian eigenvalues of model weights compared to 16-bit activations, respectively. This indicates that low-precision activation quantization also results in a flatter loss landscape.

Furthermore, these findings are not limited to standard STE-based methods. We evaluate the Hessian spectrum under other quantized training methods, including the rotation trick (Fifty et al., 2025) and QuEST (Panferov et al., 2025), and observe consistent results across 1-bit to 4-bit training. Additionally, we find that the loss Hessian in the embedding layer has most of its eigenvalues non-negative. Since most model parameters reside in the transformer layers, the saddle-point problem primarily hinders convergence. Additional evaluations of the analysis are provided in Appendix B.

### 3.2. Algorithm Design

Our findings suggest that the key to improving convergence in quantized training lies in addressing the saddle-point problem. Next, we present a fast algorithm that significantly accelerates low-precision training with minimal computational overhead.

**(1) Weight re-initialization.** Our first technique is to reset the latent weights $W$ to the linear interpolation between $W$ and $Q(W)$ during training, given a scalar $\alpha \in [0, 1]$:

$$W \leftarrow (1 - \alpha)W + \alpha Q(W). \tag{2}$$

Our motivation is that this interpolation directly reduces the distance between $W$ and $Q(W)$, thereby lowering quantization error and improving training stability (Panferov et al., 2025), without significantly altering the training loss. If the quantization grid is fixed during interpolation, such as when using learnable step sizes (Esser et al., 2020), $Q(W)$ and its corresponding loss remain unchanged after interpolation. Even when the quantization grids depend on latent weights, empirical loss variations remain negligible (within 3%) across various bit widths and interpolation coefficients.

While $Q(W)$ remains largely unchanged, the interpolation alters the loss curvature with respect to the interpolated weights, thereby affecting the loss trajectory in subsequent steps. Empirically, we observe that interpolated weights exhibit significantly larger Hessian eigenvalue magnitudes, which in turn accelerate training across language models and bit widths. We estimate the Hessian eigenvalues at interpolated weights by varying $\alpha$ from 0 to 1, using the

---

**Algorithm 1** WINQ: **W**eight (re)-**i**nitialization with **n**oise **i**njection for **Q**uantization-aware training

**Input**: Initialization $W_0 \in \mathbb{R}^d$, a quantization function $Q(\cdot)$, a language model $f_W$
**Require**: A re-initialization interval $K$, an interpolation scalar $\alpha$, standard deviation $\sigma$, number of iterations $T$ and learning rates $\eta$
**Output:** The trained latent weights $W_T$

1: **for** $i = 0, 1, \ldots, T - 1$ **do**
2:     $U_i \leftarrow$ Sample a random noise from $\mathcal{N}(0, \sigma^2 \operatorname{Id}_d)$
3:     $W_{i+1} \leftarrow W_i - \eta \nabla_{Q(W_i + U_i)} \hat{L}_{Q(W_i + U_i)}$
4:     **if** $i + 1 (\operatorname{mod} K)$ is zero **then**
5:        $W_{i+1} \leftarrow (1 - \alpha)W_{i+1} + \alpha Q(W_{i+1})$
6:     **end if**
7: **end for**

---

LLaMA-1B trained at 2-bit precision. Setting $\alpha = 0.4$ increases the maximum absolute eigenvalue by **84%** and decreases the probability mass of zero eigenvalues by **21%** compared to the original $W$. Observations across bits from 1-bit to 4-bit are consistent and described in Appendix B.

Thus, we propose to re-initialize the latent weights periodically during training. Specifically, given a scalar $\alpha \in [0, 1]$ and an interval of $K$ steps, we re-initialize the latent weights $W$ to the linear interpolation weights every $K$ steps by Equation 2. We note that the reinitialization step does not alter AdamW's state.

**(2) Noise injection.** Second, we incorporate a noise injection method in the training of latent weights. At each iteration, we inject a random Gaussian noise $U \sim \mathcal{N}(0, \sigma^2 \operatorname{Id})$ into the latent weights $W$ and compute gradients on $Q(W + U)$. This is inspired by prior work showing that performing SGD with noise-perturbed weights improves the convergence of gradient descent in non-convex optimization near saddle points (Jin et al., 2017). In experiments, we find that noise injection tends to yield smaller negative Hessian eigenvalues (i.e., larger magnitudes) and larger gradient norms, thereby speeding up quantization-aware training. Taken together, we describe our approach, named WINQ, in Algorithm 1.

**Extension to incorporate Hadamard transform.** Our approach can be applied to quantization methods that use the Hadamard transform (Panferov et al., 2025). These methods multiply the weight vector at each layer by a Hadamard matrix before applying a quantization function, thereby reducing quantization error. Given a matrix $H \in \mathbb{R}^{d \times d}$ as a block-diagonal matrix whose diagonal blocks are per-layer Hadamard matrices, we can extend the re-initialization as:

$$W \leftarrow H^\top \left((1 - \alpha)HW + \alpha Q(HW)\right), \tag{3}$$

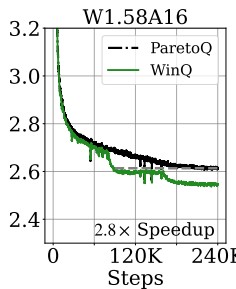 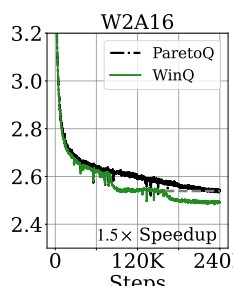 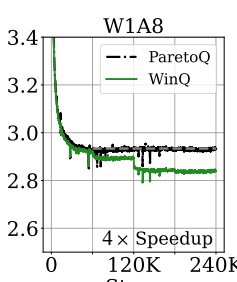 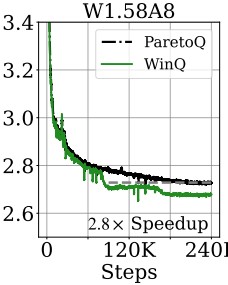 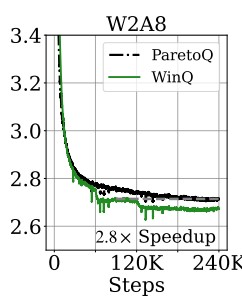

*Figure 4.* We evaluate training loss convergence by comparing our approach with the SoTA STE-based quantization-aware training method on the LLaMA-3-1B model. Our method accelerates convergence by **1.5**-**4**× across weight precisions of 1–2 bits and activation precisions of 8–16 bits. In higher-precision settings, longer re-initialization intervals generally lead to improved performance. Throughout, we use the notation W2A16 to indicate 2-bit weights and 16-bit activations, with analogous notation for other configurations. The result of W1A16 is reported in Figure 1.

based on the fact that $H$ is an orthogonal matrix. This operation can be efficiently computed with fast Hadamard multiplication kernels (Dao et al., 2025). We describe the extension with the Hadamard Transform in Appendix A.

We now describe an interpretation of weight interpolation as a proximal gradient update step on an $\ell_2$-regularized objective, which exhibits larger Hessian eigenvalues.

**Proposition 3.1.** *Let $W$ denote the continuous latent weights, $Q(W)$ the corresponding quantized weights, and $\eta > 0$ the learning rate of the optimizer. Let $q = Q(W)$ denote the quantized weights of weight $W$. Assume that the quantized weights are locally stable, meaning that there exists a neighborhood $\mathcal{N}_q \subseteq \mathbb{R}^d$ of $W$, such that $Q(Z) = q$ for all $Z \in \mathcal{N}_q$. The weight interpolation step $W \leftarrow (1 - \alpha)W + \alpha Q(W)$ with $\alpha \in [0, 1)$ is equivalent to gradient update on the $\ell_2$-regularized objective:*

$$\Phi(W) = L_Q(W) + \frac{\gamma}{2}\|W - q\|^2, \qquad (4)$$

*where $\alpha$ connects to the regularization strength $\gamma$ as $\alpha = \frac{\eta\gamma}{1+\eta\gamma}$. Correspondingly, the Hessian of $\Phi(W)$ becomes*

$$\nabla^2\Phi(W) = \nabla^2 L_Q(W) + \gamma\,\mathrm{Id}.$$

This statement requires the quantization grids to be locally constant, a condition that applies broadly to quantization methods. The proof is provided in Appendix A.3.

**Computational overhead.** Our method can be applied on top of existing quantization-aware training algorithms with negligible overhead. The re-initialization step involves simple operations like addition and multiplication. Moreover, this step is performed only every $K$ iterations, where $K$ is typically large (e.g., one-fourth of the total number of training steps), making its cost negligible compared to training. For noise injection, our approach samples a single noise vector at each step, using the same number of forward and backward passes as the base training method. We

evaluate the wall-clock time for both techniques and show that each incurs less than **1**% of the computational cost of the base training method, as detailed in Appendix B.

## 4. Experiments

In this section, we evaluate WINQ across diverse weight and activation precisions for multiple language models. First, we show that our approach accelerates the convergence of SoTA quantized training methods by up to **4**×, including the challenging setting of 1-bit weights and 8-bit activations. Second, at the same training cost, WINQ delivers up to **8.8**% improvement over leading baselines, evaluated across ten bit-width configurations and four language models. Further, we show that our approach applies to another quantization method with the Hadamard Transform, achieving up to **2.3**% gains over SoTA in the 1-bit weight and 4-bit activation setting. Finally, ablation studies confirm the contribution of both components of our approach.

### 4.1. Experimental Setup

Our experiments evaluate quantization-aware training with sub-4-bit weight quantization, including 4, 3, 2, 1.58, and 1 bit, where 1.58-bit corresponds to ternary quantization. We further combine these weight quantization settings with activation precisions of 16, 8, and 4 bits.

**Datasets and models.** We perform QAT on various language models, including LLaMA-3-1B, LLaMA-3-3B, Qwen-3-1.7B, and Qwen-3-0.6B. We perform quantized training on a language modeling dataset, FineWebEdu. We train each model up to 20B tokens, typically in 240K steps. After training, we evaluate quantized language models on several downstream tasks, following prior works. These include a language modeling dataset, WikiText2, and eight QA datasets, including ARC-easy, ARC-challenge, BoolQ, PIQA, SIQA, HellaSwag, OBQA, and WinoGrande. These are further described in Appendix B.

*Table 1.* This table reports the comparison of our approach to recent quantization methods. We evaluate quantization performance across various settings for LLaMA models, with weights quantized to 1 to 4 bits and activations to 8 to 16 bits. We report the perplexity (PPL) on WikiText2 and the average zero-shot test accuracy (Acc.) across eight QA datasets. To show relative performance, we also evaluate the full-precision (FP) pretrained model. W2A16 means 2-bit weights and 16-bit activations, and analogous notations for others. 1.58-bit means ternary quantization.

| | LLaMA-1B, W1A16 | | LLaMA-1B, W1.58A16 | | LLaMA-1B, W2A16 | | LLaMA-1B, W3A16 | | LLaMA-1B, W4A16 | |
|---|---|---|---|---|---|---|---|---|---|---|
| Metrics | PPL ($\downarrow$) | Acc. ($\uparrow$) | PPL ($\downarrow$) | Acc. ($\uparrow$) | PPL ($\downarrow$) | Acc. ($\uparrow$) | PPL ($\downarrow$) | Acc. ($\uparrow$) | PPL ($\downarrow$) | Acc. ($\uparrow$) |
| FP Model | 9.6 | 58.5 | 9.6 | 58.5 | 9.6 | 58.5 | 9.6 | 58.5 | 9.6 | 58.5 |
| RTN (PTQ) | 4.2e8 | 33.7 | 1.8e6 | 36.2 | 1.5e6 | 38.5 | 30.9 | 38.8 | 13.9 | 52.4 |
| GPTQ (PTQ) | 3.3e8 | 32.7 | 4.6e4 | 32.8 | 3.3e2 | 36.8 | 68.6 | 41.1 | 13.4 | 52.8 |
| AWQ (PTQ) | - | - | - | - | 2.0e5 | 36.4 | 1.5e2 | 42.0 | 12.2 | 56.4 |
| SpinQuant (PTQ) | 2.4e8 | 33.7 | 2.2e3 | 32.6 | 46.7 | 38.3 | 12.6 | 51.9 | 10.3 | 56.5 |
| ParetoQ | 16.9 | 51.9 | 14.0 | 54.7 | 12.5 | **56.7** | 10.9 | 57.2 | 10.3 | **58.7** |
| WINQ | **15.3** | **52.6** | **12.9** | **55.6** | **11.9** | 56.6 | **10.9** | **57.8** | **10.2** | 58.6 |

| | LLaMA-1B, W1A8 | | LLaMA-1B, W1.58A8 | | LLaMA-1B, W2A8 | | LLaMA-3B, W1A8 | | LLaMA-3B, W1.58A8 | |
|---|---|---|---|---|---|---|---|---|---|---|
| | PPL ($\downarrow$) | Acc. ($\uparrow$) | PPL ($\downarrow$) | Acc. ($\uparrow$) | PPL ($\downarrow$) | Acc. ($\uparrow$) | PPL ($\downarrow$) | Acc. ($\uparrow$) | PPL ($\downarrow$) | Acc. ($\uparrow$) |
| FP Model | 9.6 | 58.5 | 9.6 | 58.5 | 9.6 | 58.5 | 7.7 | 65.2 | 7.7 | 65.2 |
| RTN (PTQ) | 4.7e8 | 33.7 | 1.8e6 | 36.2 | 1.5e6 | 36.1 | 7.3e7 | 33.7 | 7.9e5 | 33.2 |
| GPTQ (PTQ) | 3.8e8 | 31.7 | 7.5e4 | 32.7 | 3.8e4 | 32.7 | 5.9e7 | 32.8 | 2.7e5 | 33.1 |
| SpinQuant (PTQ) | 3.4e8 | 32.8 | 5.8e3 | 32.7 | 3.8e2 | 34.9 | 4.5e7 | 32.8 | 3.1e3 | 33.3 |
| ParetoQ | 23.3 | 48.2 | 18.2 | 51.9 | 16.9 | 52.2 | 15.7 | 54.0 | 13.1 | 55.9 |
| WINQ | **21.9** | **49.0** | **16.9** | **52.5** | **16.3** | **53.0** | **14.8** | **55.2** | **12.2** | **58.6** |

**Baselines.** We compare our algorithm with existing quantization methods for language models. We mainly compare our algorithm against two state-of-the-art QAT methods, including ParetoQ (Liu et al., 2025b) and QuEST (Panferov et al., 2025). Besides, we compare with post-training quantization (PTQ) methods to illustrate the relative improvement of QAT methods. These include Round-to-Nearest (RTN), GPTQ (Frantar et al., 2023), AWQ (Lin et al., 2024), and SpinQuant (Liu et al., 2025a). We also include the full-precision result for reference.

**Implementations.** We implement our approach on top of existing QAT methods, adopting the same quantization functions. Specifically, we use Elastic Binarization (Liu et al., 2022) for 1-bit weights, Stretched Elastic Quant (Liu et al., 2025b) for 1.58 and 2-bit weights, and LSQ (Esser et al., 2020) for 3 and 4-bit weights. Activations are quantized using symmetric quantization. For comparisons with QuEST, we follow its setup, incorporating the Hadamard transform, MSE-optimal fitting, and the trust gradient estimator. As in prior methods, we quantize the non-embedding weights.

Our approach includes three key hyperparameters: the re-initialization interval $K$, the interpolation scalar $\alpha$, and the noise standard deviation $\sigma$. We vary $K$ across 40K, 60K, and 80K for a total of 240K steps; $\alpha$ between 0.1 and 0.6; and $\sigma$ between 0.0002 and 0.002. The training uses the AdamW optimizer with a learning rate in the range of $1 \times 10^{-5}$ to $4 \times 10^{-5}$. We discuss the hyperparameter tuning in Section 4.3 and report the hyperparameters used for each result in Appendix B.

### 4.2. Experimental Results

**Comparing training costs.** First, we compare the training cost of our approach with the state-of-the-art method, ParetoQ, for weights in 1–2 bits and activations in 8–16 bits. The speed-up rate is measured by the reduction in the number of training steps required to reach the same error. As shown in Figure 4, WINQ consistently outperforms ParetoQ across six bit-width settings, with larger acceleration at lower precisions. For weight-only quantization, WINQ achieves **1.5–2.8×** speed-up from 2-bit to 1-bit. With both weights and activations quantized, it delivers up to a **4×** speed-up in the challenging setting of 1-bit weights and 8-bit activations. Furthermore, WINQ converges to a lower training loss, thereby improving asymptotic performance relative to the current state of the art.

**Comparing quantization performance.** We present the quantization performance of low-precision weights (1–2 bits) and activations (8–16 bits) on LLaMA models in Table 1. Our approach advances the state of the art across eight weight- and activation-quantization settings. For weight-only quantization, WINQ surpasses the strongest baseline by up to **8.8%** in PPL and **1.6%** in average zero-shot accuracy, while also markedly outperforming all existing PTQ methods. When both weights and activations are quantized, it achieves relative gains of **7.1%** in PPL and **1.6%** in accuracy over the best baseline. On larger models such as LLaMA-3B, WINQ delivers a **6.8%** relative improvement. Consistent with the convergence results, these gains are most pronounced at lower precisions.

*Table 2.* We report the quantization performance of applying our approach to Qwen models, as compared to the SoTA method, ParetoQ. We evaluated the models in sub-2-bit weights and 8-bit activations.

| | Qwen-1.7B, W1A8 | | Qwen-0.6B, W1A8 | |
|---|---|---|---|---|
| | PPL ($\downarrow$) | Acc. ($\uparrow$) | PPL ($\downarrow$) | Acc. ($\uparrow$) |
| FP Model | 16.2 | 58.1 | 16.2 | 58.1 |
| ParetoQ | 46.5 | 42.2 | 64.0 | 41.2 |
| WINQ | **45.6** | 42.2 | **61.9** | **41.4** |
| | Qwen-1.7B, W2A8 | | Qwen-0.6B, W2A8 | |
| ParetoQ | 22.2 | 47.8 | 32.0 | 43.3 |
| WINQ | **21.8** | **48.2** | 32.0 | **43.9** |

*Table 3.* We report the quantization performance of our approach with Hadamard transform on LLaMA-1B, applied on top of another SoTA method, QuEST. We evaluate the model in sub-2-bit weights and 4-bit activations.

| | LLaMA-3-1B, W1A4 | |
|---|---|---|
| | PPL ($\downarrow$) | Acc. ($\uparrow$) |
| FP Model | 9.6 | 58.5 |
| QuEST | 42.9 | 42.4 |
| WINQ w/ Hadamard transform | **42.3** | **43.0** |
| | LLaMA-3-1B, W2A4 | |
| QuEST | 17.4 | 48.6 |
| WINQ w/ Hadamard transform | **16.9** | **49.3** |

We observe that our approach yields larger performance gains at precisions below 3 bits. At higher bit-widths, such as 4-bit, the performance of existing quantized models is close to that of the full-precision model, leaving limited room for improvement. We also measure the performance gap between the quantized and full-precision model. Our approach reduces the gap by **17%** on average compared to SoTA baselines. We include results for various bit widths for completeness, and our approach does not degrade performance at higher precision.

**Extensions.** We extend our approach to the Qwen models and observe consistent improvements over the state-of-the-art baseline. As shown in Table 2, WINQ achieves up to a **3.2%** relative reduction in PPL and a **1.3%** gain in zero-shot test accuracy compared to ParetoQ.

We further demonstrate the broad applicability of our approach to different quantization methods. When combined with another state-of-the-art method (Panferov et al., 2025) using Hadamard transform (Dao et al., 2025), WINQ improves performance by **2.8%** in 1-bit weight and 4-bit activation quantization. The results are shown in Table 3.

### 4.3. Ablation Studies

We discuss the main hyperparameters of WINQ, including the interpolation scalar $\alpha$, a re-initialization interval $K$, and the standard deviation $\sigma$ in noise injection.

**Effect of weight interpolation on training speed.** We first examine the effect of the interpolation scalar $\alpha$ in weight re-initialization. Recall that values of $\alpha$ around 0.4 tend to yield the largest increase in the maximum absolute eigenvalues of the Hessian. We vary $\alpha$ when training the LLaMA-1B model in 1-bit precision. We observe that $\alpha$ around 0.4 leads to the fastest convergence after re-initialization, corresponding to an increase in the magnitude of Hessian eigenvalues, as shown in Figure 5. We also note that the optimal $\alpha$ can vary across bit-width settings and models.

**Leaving out each step from the algorithm.** Further, we perform an ablation study that varies each hyperparameter of our approach for training the LLaMA-1B model in 1-bit precision. Both components of our approach contribute to the final performance. Results are shown in Table 4. Compared to training without noise ($\sigma = 0$), our approach improves performance by about 4%. Compared to training without weight re-initialization ($\alpha = 0$), our approach improves performance by about 6.7%. Moreover, we find that smaller values of $\alpha$ and larger re-initialization intervals tend to yield the best results.

**Strategies for selecting hyperparameters.** Lastly, we discuss the strategies for choosing the hyperparameters. Recall that interpolation is defined as $(1 - \alpha)W + \alpha Q(W)$, which reduces $\|W - Q(W)\|$ by a factor of $\alpha$. After two re-initializations, this distance is approximately $(1 - \alpha)^2$ of its original value. Smaller values of $\alpha$ keep the weights closer to the latent weights. Empirically, we find that re-initializing too close to either the latent or the quantized weights yields suboptimal performance. To balance this trade-off, we choose relatively small $\alpha$ values from 0.1 to 0.4 and perform at most three re-initializations, typically every 60K or 80K steps out of 240K steps.

For the standard deviation $\sigma$ in noise injection, we find that the optimal $\sigma$ can vary across models, with $\sigma = 0.001$ as the default for LLaMA models, and $\sigma = 0.0002$ works the best for Qwen models. We report the hyperparameters used in every experiment in Table 15 of Appendix B.

## 5. Related Work

Second-order information has provided key information for designing quantization methods (Yao et al., 2018). For example, the quantization error can be approximated using a second-order Taylor expansion (Kwun et al., 2025). OBQ (Frantar & Alistarh, 2022) and GPTQ (Frantar et al., 2023) design efficient algorithms to iteratively perform layerwise quantization using an estimated Hessian matrix.

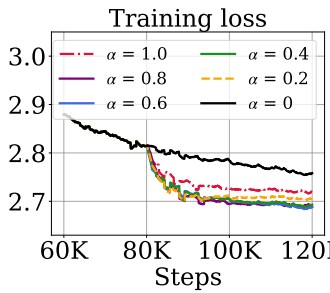

*Figure 5.* Convergence of training loss after re-initialization at 80K steps with various $\alpha$, evaluated in training LLaMA-3-1B with 1-bit weight quantization.

| PPL ($\downarrow$) | Weight re-initialization | | | Noise injection | |
|---|---|---|---|---|---|
| | $K =$40K | $K =$60K | $K =$80K | $\alpha = 0.4, K = 60$K | |
| $\alpha = 0.0$ | 16.5 | 16.5 | 16.5 | $\sigma = 0$ | 16.0 |
| $\alpha = 0.2$ | 15.6 | 15.5 | 15.5 | $\sigma = 0.0005$ | 15.7 |
| $\alpha = 0.4$ | 16.1 | **15.3** | 15.6 | $\sigma = 0.001$ | **15.3** |
| $\alpha = 0.6$ | 16.2 | 15.5 | 15.8 | $\sigma = 0.002$ | 16.3 |
| $\alpha = 0.8$ | 16.3 | 16.0 | 16.1 | $\sigma = 0.004$ | 18.5 |

*Table 4.* Results from varying hyperparameters including the interpolation scalar $\alpha$, re-initialization interval $K$, and standard deviation $\sigma$. We train LLaMA-3-1B with 1-bit weight quantization and report the perplexity (PPL) on WikiText-2. $\sigma = 0$: no noise injection. $\alpha = 0$: no weight interpolation.

QuIP (Chee et al., 2023) proposes incoherence processing that minimizes a proxy objective derived from the second-order approximation through adaptive rounding. In these works, the loss Hessian is typically estimated from the second-order moments of input features. By computing Hessian top eigenvalues with Hessian-vector products (Yao et al., 2020), HAWQ (Dong et al., 2019) proposes a mixed-precision quantization method that uses the top eigenvalues as the layerwise sensitivity measure. Further, using the Hessian trace as a sensitivity measure has improved mixed-precision performance (Dong et al., 2020). In contrast, our work examines the eigenvalue distribution of the Hessian during quantized training, revealing behavior distinct from that observed in other deep learning settings (Yao et al., 2018). Our analysis suggests that the key challenge in quantization-aware training is addressing the saddle-point problem. In particular, Hessian spectral statistics are closely connected to the generalization and influence estimation of deep neural networks (Ju et al., 2022; Zhang et al., 2024a; 2025; 2026b). See a recent review article for further discussions from a PAC-Bayes data-dependent perspective (Wilson, 2025).

Quantized training has been widely used for low-bit quantization (Yin et al., 2024). It has also been applied to train a gated quantized variational autoencoder to design a learnable tokenizer in LLMs (Datta et al., 2025). One focus of prior work is improving gradient estimation in low-precision settings. Quant-Noise (Fan et al., 2021) proposes to only quantize a random subset of weights at each training iteration. Fifty et al. (2025) propose a rotation trick that rotates and linearly rescales the gradients to improve gradient estimation for vector quantization. For further references, see the survey of Gholami et al. (2022).

Another focus in quantization-aware training is to design alternative formulations with regularization to improve stability and establish convergence guarantees. ProxQuant (Bai et al., 2019) reformulates a regularized learning problem via proximal gradient methods that apply a prox op-

erator in between stochastic gradient steps on the latent weight. LOTION (Kwun et al., 2025) proposes a smoothing framework that reformulates quantized loss as the expectation of the loss under randomized-rounding noise, which can be interpreted as a second-order curvature-aware regularizer. CAGE (Tabesh et al., 2025) proposes a multi-objective formulation to balance the training loss with the distance between latent and quantized weights, regularizing the weights toward the quantization grid. In contrast, we analyze quantized training by examining second-order gradients and connecting convergence to saddle-point problems. Furthermore, our weight interpolation technique can be interpreted as a proximal update that regularizes the distance between latent and quantized weights.

An active line of research has focused on designing optimizers to accelerate model training. One approach leverages layerwise adaptive learning rates with large batch sizes (You et al., 2020). Another direction is to improve the preconditioner design in Adam, as in Muon or its variants. Recent empirical analyses show that using the full Gauss-Newton matrix as a preconditioner substantially outperforms optimizers that estimate the second-order structure (Abreu et al., 2026).

## 6. Conclusion

This work studies the slow convergence in quantization-aware training for language models. Our analysis of the loss Hessian revealed that model weights tend to get stuck in flat regions near saddle points, especially at lower precision. This insight led us to develop a simple method to accelerate quantization-aware training by combining weight reinitialization with noise injection. Our approach demonstrates significant speedup and improved quantization performance compared to various methods. These findings provide an improved understanding of quantization-aware training from an optimization perspective and open up many new directions for designing scalable training methods for LLMs in low-precision.

## Acknowledgement

The majority of this work was completed during Dongyue Li's internship at Meta. The work of Dongyue Li, Zhenshuo Zhang, and Hongyang Zhang is also partially supported by NSF award IIS-2412008 and a startup grant from Northeastern University. Any views or opinions expressed herein are solely those of the authors listed, and may differ from those expressed by the National Science Foundation.

## Impact Statement

Our paper focuses on quantization-aware training. The goal is to advance the field of machine learning. There may be potential societal consequences of our work, none of which we feel must be specifically highlighted here.

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

# A. Omitted Details of Our Approach

## A.1. Notations and Terminologies

To precisely describe the concepts used in this paper, we introduce the notations and definitions following the terminology from (Jin et al., 2017). For a function $f : \mathbb{R}^d \to \mathbb{R}$, we use $\nabla f(\cdot)$ and $\nabla^2 f(\cdot)$ to denote its gradient and Hessian. For vectors we use $\| \cdot \|$ to denote the $\ell_2$-norm. We use $\lambda_{\max}(\cdot), \lambda_{\min}(\cdot), \lambda_i(\cdot)$ denote the maximum, minimum, and $i$-th eigenvalues.

We begin by formalizing the smoothness property of functions, which ensures that the gradient does not change too rapidly.

**Definition A.1** ($C$-smooth). A differentiable function $f : \mathbb{R}^d \to \mathbb{R}$ is said to be $C$-smooth (or $C$-gradient Lipschitz) if

$$\|\nabla f(x_1) - \nabla f(x_2)\| \leq C \|x_1 - x_2\|, \; \forall\, x_1 \in \mathbb{R}^d, x_2 \in \mathbb{R}^d.$$

Then, we define stationary points, which are critical in analyzing the convergence of gradient descent methods.

**Definition A.2** (First-order stationary point). For a differentiable function $f(\cdot)$, we say that $x$ is a *first-order stationary point* if $\|\nabla f(x)\| = 0$. We also say $x$ is an *$\varepsilon$-first-order stationary point* if

$$\|\nabla f(x)\| \leq \varepsilon.$$

A first-order stationary point can be either a local minimum, a saddle point, or a local maximum. We define local minima as follows.

**Definition A.3** (Local minimum). For a differentiable function $f(\cdot)$, a point $x$ is a *local minimum* if it is a first-order stationary point and there exists $\delta > 0$ such that

$$f(x) \leq f(y), \; \forall\, y \in B(x, \delta),$$

where $B(x, \delta)$ denotes the ball of radius $\delta$ centered at $x$.

Finally, not all stationary points are desirable in optimization. In particular, a point may be stationary but not a local minimum. We define saddle points as follows.

**Definition A.4** (Saddle point). For a differentiable function $f(\cdot)$, a point $x$ is a *saddle point* if it is a first-order stationary point but not a local minimum. For a twice-differentiable function $f(\cdot)$, a saddle point $x$ is *strict* (or non-degenerate) if $\lambda_{\min}(\nabla^2 f(x)) < 0$.

In our evaluations, we observe that the model weights in quantization-aware training converge to an approximate first-order stationary point, while the corresponding loss Hessian exhibits more than one negative eigenvalue. This indicates that the model weights are around saddle points.

Additionally, near a stationary point, the local convergence rate of gradient-based methods is strictly governed by the eigenvalue spectrum of the loss Hessian (Nesterov, 2018). Therefore, we measure the Hessian spectrum and the Hessian eigenvalues with the largest absolute values to monitor the convergence of quantized training.

## A.2. Estimating the Hessian spectrum

In this section, we briefly describe the Stochastic Lanczos Quadrature (SLQ) algorithm (Bai et al., 1996), which we use to estimate the empirical eigenvalue distribution of the loss Hessian. The method samples the spectrum using random probe vectors and computes the spectrum via the Lanczos algorithm, reducing it to a small tridiagonal matrix whose eigenvalues approximate those of the Hessian.

Let $H$ denote a large symmetric matrix (e.g., the Hessian of a loss function with respect to model weights). The spectral density is defined as:

$$p(\lambda) = \frac{1}{n} \sum_{i=1}^{n} \delta(\lambda - \lambda_i),$$

where $\lambda_i$ are the eigenvalues of $H$ and $\delta$ denotes the Dirac delta function. Since computing the Hessian matrix is infeasible in large language models, the method uses Hessian–vector products to estimate the Hessian spectrum.

---

**Algorithm 2** Stochastic Lanczos Quadrature (SLQ) for estimating the Hessian spectrum

---

1:  **Input:** A symmetric Hessian operator $H \in \mathbb{R}^{d \times d}$, accessed only through Hessian–vector products
2:  **Require:** Number of probe vectors $m$, number of Lanczos steps $k$
3:  **for** $j = 1, 2, \ldots, m$ **do**
4:      Sample a Rademacher vector $v^{(j)} \in \{-1, +1\}^d$ and set $q_1 \leftarrow v^{(j)} / \|v^{(j)}\|_2$
5:      Initialize $q_0 \leftarrow 0$ and $\beta_0 \leftarrow 0$
6:      **for** $i = 1, 2, \ldots, k$ **do**
7:          $z \leftarrow H q_i - \beta_{i-1} q_{i-1}$
8:          $\alpha_i \leftarrow q_i^\top z$
9:          $\beta_i \leftarrow \|z - \alpha_i q_i\|_2$
10:          $q_{i+1} \leftarrow (z - \alpha_i q_i)/\beta_i$
11:      **end for**
12:      Form the tridiagonal matrix $T_j \in \mathbb{R}^{k \times k}$ with diagonal $(\alpha_1, \ldots, \alpha_k)$ and off-diagonal $(\beta_1, \ldots, \beta_{k-1})$
13:      Compute the eigendecomposition $T_j = U_j \text{diag}(\theta_1^{(j)}, \ldots, \theta_k^{(j)}) U_j^\top$
14:      Set $w_i^{(j)} \leftarrow \left(U_j[1, i]\right)^2$ for $i = 1, \ldots, k$
15:  **end for**
16:  **Return** A set $\{(\theta_i^{(j)}, w_i^{(j)})\}_{i=1,\ldots,k;j=1,\ldots,m}$ that approximate the spectral density as in Equation 5

---

The algorithm works by framing the spectral density as a matrix trace, which can be approximated via Hutchinson's method using random probe vectors. Specifically, for a given random vector, SLQ applies $k$ iterations of the Lanczos algorithm to compress the local curvature information into a small tridiagonal matrix. Diagonalizing this much smaller matrix yields a set of approximated eigenvalues $\{\theta_i\}$ and corresponding weights $\{w_i\}$.

By repeating this procedure across $m$ random Rademacher vectors $v_j$, we obtain $m$ sets of eigenvalues $\theta_i^{(j)}$ and weights $w_i^{(j)}$. These are aggregated to form the final approximation of the overall Hessian spectrum:

$$\frac{1}{m} \sum_{j=1}^{m} \sum_{i=1}^{k} w_i^{(j)} \delta(\lambda - \theta_i^{(j)}). \tag{5}$$

Finally, one can convolve this discrete approximation with a Gaussian kernel to obtain a smooth, continuous estimate of the spectral density. We summarize the procedure in Algorithm 2.

In our evaluation, we estimate the empirical eigenvalue distribution of the loss Hessian using the model's loss on $2 \times 10^5$ tokens from the training dataset. We apply the SLQ method with $m = 200$ random vectors and $k = 100$ steps. Then, we estimate the empirical distribution obtained from the $\{\theta_i^{(j)}, w_i^{(j)}\}$, where the weights are normalized to ensure the total probability mass equals one. We report the resulting discrete distribution without additional kernel smoothing. The Hessian-vector products are implemented using existing PyTorch functions.

### A.3. Theoretical Connection

We now provide a theoretical interpretation of the weight interpolation step through the lens of proximal optimization (Bai et al., 2019). We show that the weight interpolation step is equivalent to a proximal update step in a regularized training objective.

*Proof of Proposition 3.1.* We consider optimizing the objective $\Phi(W)$ using proximal gradient descent. We decompose the objective into the loss component $L_Q(W)$, which is optimized via the Straight-Through Estimator, and the regularization component $R(W) = \frac{\gamma}{2} \|W - q\|_2^2$, where $q = Q(W)$ is the target quantization grid point. We assume $q$ is locally constant within the current quantization cell.

A proximal gradient descent update at iteration $t$ consists of two components:

- Gradient update step: we conduct a gradient step on the loss $L_Q(W)$ using the learning rate $\eta$:

$$V = W_t - \eta \nabla L_Q(W_t).$$

---

**Algorithm 3** WINQ with the Hadamard Transform

---

**Input**: Initialization $W_0 \in \mathbb{R}^d$, a quantization function $Q(\cdot)$, a language model $f_W$, a matrix $H$ of $d$ dimension for the Hadamard Transform

**Require**: A re-initialization interval $K$, an interpolation scalar $\alpha$, standard deviation $\sigma$, the number of iterations $T$ and learning rates $\eta$

**Output:** The trained latent weights $W_T$

1: **for** $i = 0, 1, \ldots, T - 1$ **do**
2:     $U_i \leftarrow$ Sample a random noise from $\mathcal{N}(0, \sigma^2 \,\mathrm{Id}_d)$
3:     $W_{i+1} \leftarrow W_i - \eta H^\top \nabla_{Q(H(W_i+U_i))} \hat{L}_{Q(H(W_i+U_i))}$   ▷ The HT is also applied to activations
4:     **if** $i + 1 (\mod K)$ is zero **then**
5:        $W_{i+1} \leftarrow H^\top \left( (1 - \alpha) H W_{i+1} + \alpha Q(H W_{i+1}) \right)$
6:     **end if**
7: **end for**

---

- Proximal update step: we apply the proximal operator of the scaled regularizer $\eta R(W)$ to the intermediate weight $V$:

$$W_{t+1} = \mathrm{prox}_{\eta R}(V) = \arg \min_W \left( R(W) + \frac{1}{2\eta} \|W - V\|_2^2 \right).$$

Substituting the $\ell_2$ penalty $R(W)$ into the proximal operator yields:

$$W_{t+1} = \arg \min_W \left( \frac{\gamma}{2} \|W - q\|_2^2 + \frac{1}{2\eta} \|W - V\|_2^2 \right).$$

As the objective is a convex quadratic function, this has a unique global minimum where

$$\nabla_W \left( \frac{\gamma}{2} \|W - q\|_2^2 + \frac{1}{2\eta} \|W - V\|_2^2 \right) = \gamma (W - q) + \frac{1}{\eta} (W - V) = 0$$

Therefore, we have

$$W = \left( \frac{1}{1 + \eta\gamma} \right) V + \left( \frac{\eta\gamma}{1 + \eta\gamma} \right) q.$$

When setting the interpolation scalar $\alpha$ as $\frac{\eta\gamma}{1+\eta\gamma}$. Substituting $\alpha$ and $1 - \alpha$ back into our equation yields the exact weight interpolation step:

$$W_{t+1} = (1 - \alpha) V + \alpha q.$$

This shows that the weight interpolation is mathematically equivalent to a proximal update applied to the $\ell_2$ penalty. Additionally, the second derivative of the quadratic penalty term is $\gamma I$, which yields

$$\nabla^2 \Phi(W) = \nabla^2 L_Q(W) + \gamma \,\mathrm{Id}.$$

This, in turn, increases the magnitude of the Hessian eigenvalues.       □

### A.4. Extensions

Next, we extend our approach to incorporate the Hadamard Transform for quantization-aware training. In this method, a Hadamard matrix is applied to weights and activations before quantization at each layer. We denote by $H$ a block-diagonal matrix whose diagonal blocks are the Hadamard matrices of each layer. To integrate it, we modify the re-initialization step by multiplying the interpolated weights by the inverse Hadamard matrix, which can be efficiently computed using the Hadamard matrix's transpose. The procedure is described in Algorithm 3.

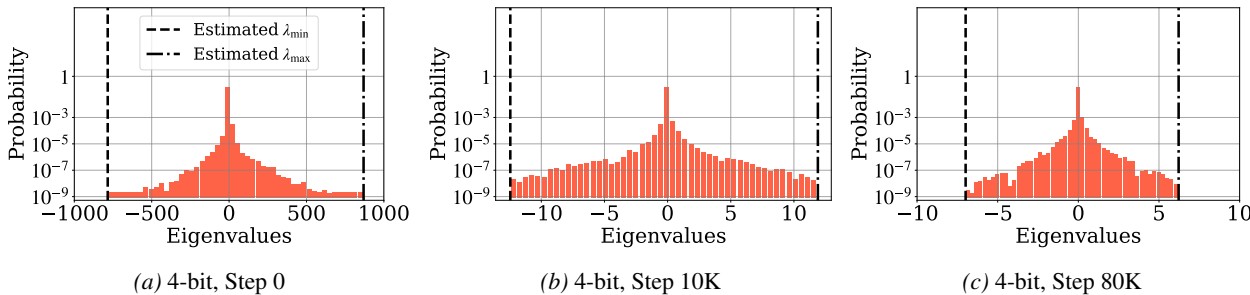

*(a)* 4-bit, Step 0       *(b)* 4-bit, Step 10K       *(c)* 4-bit, Step 80K

*Figure 6.* We plot the estimated Hessian eigenvalue distribution of the model weights at different steps of 4-bit QAT training. The $x$-axis represents the eigenvalues, and the $y$-axis shows their probability mass on a log scale.

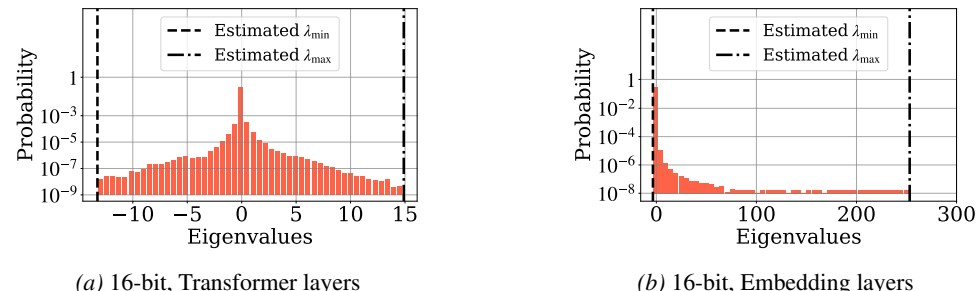

*(a)* 16-bit, Transformer layers       *(b)* 16-bit, Embedding layers

*Figure 7.* We plot the estimated Hessian eigenvalue distribution of the full-precision pretrained model. The eigenvalues have larger magnitudes compared to those in lower-precision settings. The $x$-axis represents the eigenvalues, and the $y$-axis shows their probability mass on a log scale.

## B. Omitted Experiments

### B.1. Implementation

Specifically, we use Elastic Binarization (Liu et al., 2022) for 1-bit weights, Stretched Elastic Quant (Liu et al., 2025b) for 1.58 and 2-bit weights, and LSQ (Esser et al., 2020) for 3 and 4-bit weights. Activations are quantized using symmetric quantization. For comparisons with QuEST, we follow its setup, incorporating the Hadamard Transform, MSE-optimal fitting, and the trust gradient estimator. As in prior methods, we quantize the non-embedding weights.

To facilitate the reproduction of the experimental results presented in this paper, we provide a detailed description of the algorithmic procedure in Section 3. The datasets, models, and hyperparameter-tuning strategies used in our experiments are discussed in Section 4. All datasets and models used in this work are publicly accessible online. Their respective sources are documented in Table 10 of Appendix B, and the exact hyperparameters corresponding to each reported result are provided in Table 15 of Appendix B.

### B.2. Evaluation of Hessian Spectrum

Figure 6 shows the empirical eigenvalue distributions of the loss Hessian during 4-bit quantization-aware training. The results align with those observed for 3-bit quantization. We find that most eigenvalues cluster near zero, and after 80K training steps, the maximum absolute eigenvalue falls below 10.

Second, we illustrate the eigenvalue distribution of the weights in the transformer layers and the embedding layer, using the full-precision pretrained model. As shown in Figure 7, we observe that in the transformer layers, the Hessian shows a balanced mix of negative and positive eigenvalues, with larger magnitudes at lower precision, whereas the embedding layer consistently exhibits non-negative eigenvalues.

Third, we present the empirical loss Hessian eigenvalue distributions for the embedding layer weights in Figure 8. Following common practice, we do not quantize the embedding layer. Interestingly, after training, its eigenvalues remain non-negative, unlike those of the transformer layers. Moreover, we observe that, at lower bit precisions, both the maximum eigenvalue and the trace (sum of eigenvalues) decrease.

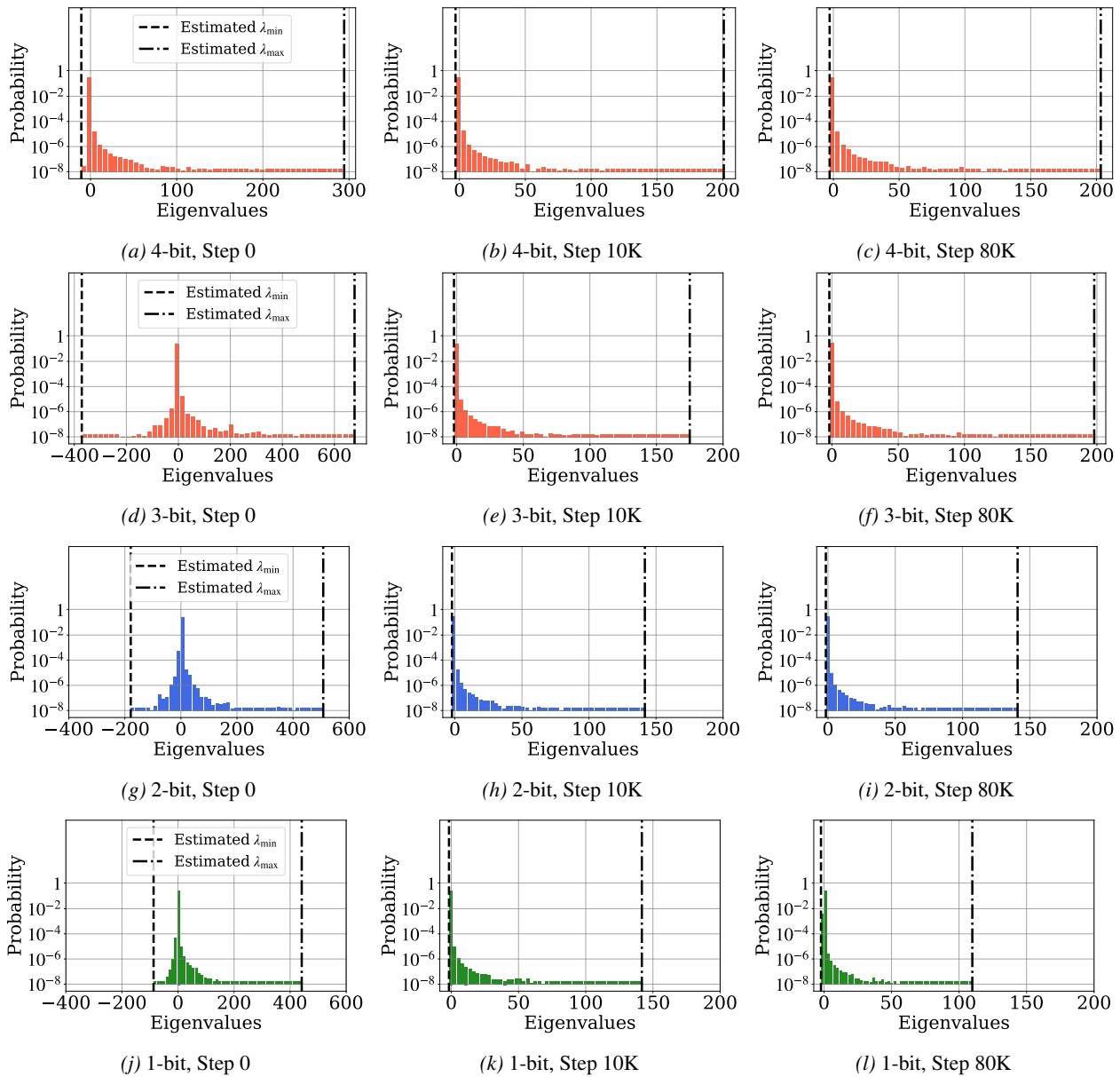

*Figure 8.* We plot the estimated Hessian eigenvalue distribution of the weights in the embedding layer. We illustrate the distribution for quantization-aware training with 4-bit, 3-bit, 2-bit, and 1-bit quantization at various training steps. The $x$-axis represents the eigenvalues, and the $y$-axis shows their probability mass on a log scale.

Our observations are not limited to standard STE-based methods. We further evaluate other gradient estimation methods, including the rotation trick (Fifty et al., 2025) and QuEST (Panferov et al., 2025). We train a 30M LLaMA-style model on SlimPajama with 1–4 bit weight quantization, following the setup in QuEST (Panferov et al., 2025).

Across these methods, we observe consistent results. As training progresses, Hessian eigenvalues increasingly concentrate near zero, and their magnitudes decrease substantially (from around 180 at 1K steps to 3 at 10K steps in 1-bit quantized training). Moreover, lower bit-widths exhibit smaller eigenvalue magnitudes (from 15 in 3-bit to 3 in 1-bit). These results suggest that the phenomenon is consistent in quantization-aware training.

Recall that the magnitude of Hessian eigenvalues increases when evaluating interpolated weights between $W$ and $Q(W)$, which motivates our weight re-initialization technique for accelerating training.

*Table 5.* We report the maximum absolute Hessian eigenvalues, $\max_i \|\lambda_i\|$, evaluated at interpolated weights $(1 - \alpha)W + \alpha Q(W)$ for Llama-1B trained with 1–4 bit weight quantization and 16-bit activations. Across all bit-widths, curvature consistently increases with interpolation and typically peaks around $\alpha = 0.4$. We compute the Hessian eigenvalues with respect to the parameters, including the learnable quantization step sizes.

| $\max_i \|\lambda_i\|$ | $\alpha = 0.0$ | $\alpha = 0.2$ | $\alpha = 0.4$ | $\alpha = 0.6$ | $\alpha = 0.8$ | $\alpha = 1.0$ |
|---|---|---|---|---|---|---|
| 1-bit QAT | $0.93 \pm 0.3$ | $1.20 \pm 0.1$ | $1.72 \pm 0.3$ | $\mathbf{1.83} \pm 0.1$ | $1.24 \pm 0.1$ | $1.13 \pm 0.1$ |
| 2-bit QAT | $1.64 \pm 0.5$ | $2.45 \pm 0.4$ | $\mathbf{3.09} \pm 0.4$ | $2.65 \pm 0.5$ | $2.42 \pm 0.5$ | $2.05 \pm 0.5$ |
| 3-bit QAT | $6.24 \pm 0.4$ | $6.79 \pm 0.4$ | $\mathbf{7.70} \pm 0.3$ | $6.44 \pm 0.3$ | $6.68 \pm 0.3$ | $6.38 \pm 0.2$ |
| 4-bit QAT | $7.13 \pm 0.4$ | $7.31 \pm 0.4$ | $\mathbf{8.06} \pm 0.5$ | $7.61 \pm 0.5$ | $7.66 \pm 0.5$ | $7.12 \pm 0.6$ |

*Table 6.* We report the relative gradient norms and maximum absolute Hessian eigenvalues evaluated at interpolated weights for Llama-1B trained with 2-bit quantization and 16-bit activations under varying noise standard deviation $\sigma$. Noise injection tends to yield smaller negative Hessian eigenvalues (i.e., larger absolute values) compared to standard training.

| | $\hat{L}_{Q(W)}$ | $\|\nabla_W \hat{L}_{Q(W)}\|_2 / \|W\|_2$ | $\max_i \|\lambda_i\|$ | | | | | |
|---|---|---|---|---|---|---|---|---|
| | | | $\alpha = 0.0$ | $\alpha = 0.2$ | $\alpha = 0.4$ | $\alpha = 0.6$ | $\alpha = 0.8$ | $\alpha = 1.0$ |
| $\sigma = 0$ | $3.18 \pm 0.29$ | $0.011 \pm 0.003$ | $1.64 \pm 0.5$ | $2.45 \pm 0.4$ | $\mathbf{3.09} \pm 0.4$ | $2.65 \pm 0.5$ | $2.42 \pm 0.5$ | $2.05 \pm 0.5$ |
| $\sigma = 0.0005$ | $3.08 \pm 0.28$ | $0.012 \pm 0.002$ | $3.03 \pm 0.2$ | $3.53 \pm 0.2$ | $\mathbf{3.72} \pm 0.1$ | $3.50 \pm 0.2$ | $3.40 \pm 0.2$ | $3.18 \pm 0.2$ |
| $\sigma = 0.001$ | $3.12 \pm 0.29$ | $0.013 \pm 0.002$ | $3.45 \pm 0.3$ | $3.91 \pm 0.2$ | $3.95 \pm 0.2$ | $\mathbf{3.96} \pm 0.2$ | $3.55 \pm 0.2$ | $3.32 \pm 0.7$ |
| $\sigma = 0.002$ | $3.13 \pm 0.28$ | $0.013 \pm 0.003$ | $3.53 \pm 0.2$ | $3.94 \pm 0.1$ | $3.74 \pm 0.1$ | $\mathbf{3.85} \pm 0.2$ | $3.49 \pm 0.2$ | $3.53 \pm 0.2$ |

*Table 7.* We report the perplexity evaluated on the WikiText2 dataset. We train Llama-1B with 1-bit weight quantization under different noise levels $\sigma$ and batch sizes. We observe that larger batch sizes benefit from using a larger $\sigma$ (around 0.001).

| | $\sigma = 0.0005$ | $\sigma = 0.001$ | $\sigma = 0.002$ | $\sigma = 0.004$ |
|---|---|---|---|---|
| Batch Size 64 | 15.8 | **15.3** | 16.3 | 18.5 |
| Batch Size 32 | 16.2 | 16.1 | 17.0 | 19.1 |
| Batch Size 16 | 16.6 | 16.7 | 18.2 | 20.5 |

To further support this observation, we measure Hessian eigenvalue magnitudes at different interpolation points across various quantized training settings. Using Llama-1B trained with 1–4 bit quantization at 80K steps, we compute the maximum absolute Hessian eigenvalues for interpolated weights $(1 - \alpha)W + \alpha Q(W)$ over $\alpha$ between 0.0 and 1.0. Results are reported in Table 5. We find that interpolation consistently increases the curvature, with the largest magnitudes typically occurring around $\alpha = 0.4$.

Further, we evaluate Hessian eigenvalues under noise injection. Using Llama-1B trained with 2-bit weight quantization at 80K steps, we measure the maximum absolute Hessian eigenvalues for noise standard deviations $\sigma$ between 0.0005, 0.001, and 0.002. We find that models trained with noise injection exhibit smaller negative eigenvalues (i.e., larger-magnitude curvature) than those trained without noise. In addition, noise injection produces slightly larger gradient norms. The observations explain the faster training in noise injection.

### B.3. Ablation Studies

We conducted an ablation study jointly varying the batch size and the standard deviation $\sigma$ of the noise. We vary the batch size among 64, 32, and 16, and vary $\sigma$ from 0.0005 to 0.004. We perform the experiments using Llama-1B trained with 1-bit weight quantization and 16-bit activations for 240K steps. The results are shown in Table 7. We observe that when using smaller batch sizes (e.g., 16), smaller values of $\sigma$ tend to be better. For larger batches, using a larger $\sigma$ is better (typically around 0.001). A batch size of 64 yields the best overall performance. Larger batches exceed the memory limits. Accordingly, we use a batch size of 64 in the paper.

**Increasing the training budget.** In our main experiments, we follow the setup of prior work (Liu et al., 2025b) and train for 20B tokens (240K steps), using the same budget for our method. To further assess the scaling with respect to training budget, we doubled the budget to 40B tokens (480K steps) for Llama-1B with 1-, 2-, and 3-bit weights and 16-bit activations. As shown in Table 8, our approach continues to outperform ParetoQ at this larger budget, achieving an average perplexity improvement of **7%**.

*Table 8.* We report the perplexity evaluated on the WikiText2 dataset, when increasing the training budget to 40B tokens. We compare WINQ with ParetoQ under settings of W1A16, W2A16, and W3A16. WINQ consistently outperforms the baseline after the increase of the training budget. W1A16 means 1-bit weights and 16-bit activations, and analogous notations for others.

| | W1A16 | | W2A16 | | W3A16 | |
|---|---|---|---|---|---|---|
| Training budget (in the number of tokens) | 20B | 40B | 20B | 40B | 20B | 40B |
| ParetoQ | 16.9 | 16.2 | 12.5 | 12.3 | 14.0 | 13.9 |
| WINQ | **15.3** | **14.7** | **11.9** | **11.8** | **12.9** | **12.8** |

*Table 9.* We report the results of training Llama-8B with 2-bit weight and 16-bit activation quantization. We report the perplexity (PPL) on WikiText2 and the average zero-shot test accuracy across eight QA datasets.

| | PPL ($\downarrow$) | Avg. Accuracy ($\uparrow$) | ARC-e | ARC-c | BoolQ | PIQA | SIQA | HellaSwag | OBQA | WinoGrande |
|---|---|---|---|---|---|---|---|---|---|---|
| ParetoQ | 8.4 | 64.7 | 75.9 | 50.2 | 75.0 | 78.5 | 48.5 | 72.3 | 49.6 | 68.0 |
| WINQ | 8.3 | **65.3** | 76.2 | 52.0 | 76.9 | 78.4 | 48.5 | 72.7 | 50.4 | 67.7 |

*Table 10.* Below we report the source links for all the open-source language models and the text datasets used throughout our experiments.

| Model | Source |
|---|---|
| LLaMA-3-1B | https://huggingface.co/meta-LLaMA/LLaMA-3.2-1B |
| LLaMA-3-3B | https://huggingface.co/meta-LLaMA/LLaMA-3.2-3B |
| Qwen-3-1.7B | https://huggingface.co/Qwen/Qwen3-1.7B |
| Qwen-3-0.6B | https://huggingface.co/Qwen/Qwen3-0.6B |

| Dataset | Source |
|---|---|
| FineWebEdu | https://huggingface.co/datasets/HuggingFaceFW/fineweb-edu |
| Wiki2 | https://huggingface.co/datasets/Salesforce/wikitext |
| ARC-e | https://huggingface.co/datasets/mib-bench/arc_easy |
| ARC-c | https://huggingface.co/datasets/ibragim-bad/arc_challenge |
| BoolQ | https://huggingface.co/datasets/google/boolq |
| PIQA | https://huggingface.co/datasets/ybisk/piqa |
| SIQA | https://huggingface.co/datasets/lighteval/siqa |
| HellaSwag | https://huggingface.co/datasets/Rowan/hellaswag |
| OBQA | https://huggingface.co/datasets/allenai/openbookqa |
| WinoGrande | https://huggingface.co/datasets/allenai/winogrande |

### B.4. Omitted Results

Our approach can be applied directly to larger architectures. To explore the scalability of model sizes, we conducted an experiment on Llama-8B, training it with 2-bit weight quantization and 16-bit activations for 150K steps. As shown in Table 9, our approach achieves around a relative **1**% improvement in the average zero-shot accuracy over the state-of-the-art baseline (Liu et al., 2025b).

**Low-bit activation quantization.** Our approach is broadly applicable across diverse quantization settings, including extremely low-bit activation quantization. To demonstrate this, we integrate our method with QuEST (Panferov et al., 2025) in a 1-bit weight and 1-bit activation setting. Training a 30M LLaMA-style model from scratch on SlimPajama for 2500 steps, we achieve a 5% reduction in test perplexity and a 2% reduction in test loss compared to the QuEST baseline.

Furthermore, we compare our approach against CAGE (Tabesh et al., 2025), a recent work that similarly regularizes weights toward the quantization grid. While CAGE achieves this by modifying the backward update, we instead regularize the loss Hessian via weight interpolation and noise injection. Using a 4-bit weight and activation setting and training a 30M LLaMA-style model from scratch on SlimPajama for 2500 steps, our method yields an additional 1% reduction in test perplexity over CAGE.

**Full comparison results.** We describe the sources for the models and datasets in Table 10. For the weight-only quantization, we describe the results in Table 13. For weight and activation quantization on Llama models, we describe the results

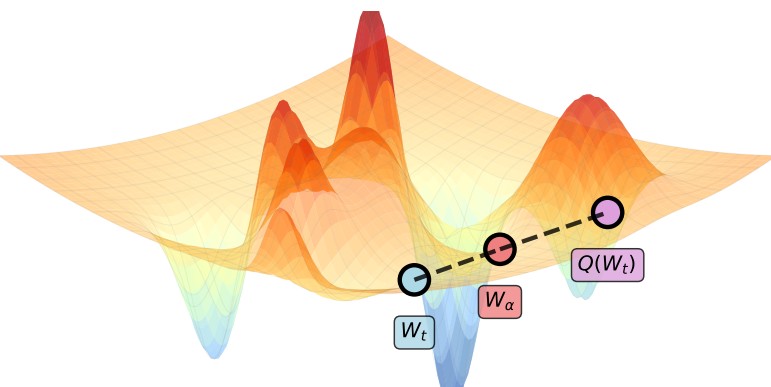

*Figure 9.* We illustrate an example of the loss surface with critical points that lie on a linear line. $W_t$ lies in a flat region of the surface, where the curvature is small in both directions, and training progress stalls. By interpolating between $W_t$ and its quantized counterpart $Q(W_t)$, the re-initialization moves the weight to $W_\alpha$ in a sharper region with larger Hessian eigenvalues. This transition increases the Hessian eigenvalues, thereby accelerating training.

in Table 14. Regarding quantization performance for Qwen models, we present the results in Table 16. For quantization performance on QuEST models, we describe the results in Table 12.

### B.5. Computational Overhead

The additional runtime introduced by WINQ is negligible relative to the base training cost. We measured wall-clock time in seconds on a machine with 26 CPUs and one A100 GPU. Reinitialization involves only element-wise addition of two weight matrices. On Llama-1B, we perform re-initialization up to three times; This takes up to 0.3 seconds, compared to 28.7 hours for the full training run. For noise injection, each iteration samples Gaussian noise and adds it element-wise to the weights. On Llama-1B, this costs 0.004 seconds per iteration, which is less than 1% of the time for a forward-backward pass (0.43 seconds).

Our approach introduces no additional memory overhead. On Llama-1B, our approach uses the same peak GPU memory as the ParetoQ baseline: 78.8 GB. This is because memory consumption is dominated by the forward and backward passes, in which our techniques do not increase memory usage.

### B.6. Loss Landscape Visualization

To further illustrate our re-initialization technique, we present a visual example of the loss surface in Figure 9. $W_t$ are located near a saddle point of the loss surface, where the curvature along two directions is close to zero, leading to stagnated training progress. In contrast, by interpolating between $W_t$ and the quantized weights $Q(W_t)$, the reinitialized weight $W_\alpha$ is reset to a region with larger Hessian eigenvalues that facilitates faster training.

### B.7. Further Extensions

In this section, we extend our study to training a model that can be quantized to multiple precisions. Specifically, given $n$ quantization functions $Q_1, Q_2, \ldots, Q_n$ (e.g., 1-bit to 4-bit quantization), our goal is to train a single model that jointly minimizes the loss across all $n$ functions. Let $L_{Q_i(W)}$ denote the expected loss of the model quantized by the $i$-th function. Our objective is then the average test loss across all quantization functions: $\frac{1}{n} \sum_{i=1}^{n} L_{Q_i(W)}$.

The motivation for this setting stems from the high computational cost of quantization-aware training, which often requires retraining separately for each quantization function. A more practical alternative is to train the model once to support arbitrary bit-width quantization, enabling efficient deployment across diverse scenarios while preserving performance comparable to that of single-bit quantization-aware training. Such an approach would enable the creation of language models at multiple precisions, which could serve as flexible foundations for further fine-tuning on downstream tasks, for example, by using high-precision low-rank adapters (Yin et al., 2024). Moreover, combining low-precision language models with KV cache quantization (e.g., (Zhang et al., 2024b)) offers a promising path toward building highly efficient inference systems for state-of-the-art LLMs.

*Table 11.* We report preliminary results on training a single model across multiple bit-widths (4, 3, 2, and 1 bit). Using LLaMA-3-1B as the base model, we follow the setup of ParetoQ. For comparison, we adopt Matryoshka Quantization (Nair et al., 2025), a recent method for multi-bit training. In these experiments, activations are fixed to 16 bits. We evaluate performance using perplexity on WikiText-2 and average zero-shot accuracy across eight QA tasks.

| LLaMA-3-1B | Wiki2 ($\downarrow$) | ARC-e | ARC-c | BoolQ | PIQA | SIQA | HellaSwag | OBQA | WinoGrande | Avg. Accuracy ($\uparrow$) |
|---|---|---|---|---|---|---|---|---|---|---|
| Single bit-width training (ParetoQ), with 640K training steps | | | | | | | | | | |
| 1-bit | 16.9 | 57.3 | 36.2 | 62.4 | 69.1 | 41.9 | 48.3 | 45.1 | 55.0 | 51.9 |
| 2-bit | 12.5 | 64.8 | 41.7 | 62.8 | 73.1 | 44.0 | 56.6 | 52.0 | 58.5 | 56.7 |
| 3-bit | 10.9 | 65.3 | 41.9 | 64.2 | 73.8 | 43.9 | 61.3 | 47.7 | 59.5 | 57.2 |
| 4-bit | 10.3 | 67.4 | 43.4 | 64.4 | 74.8 | 44.4 | 63.5 | 50.4 | 61.4 | 58.7 |
| Multi-level bit-width training (Matryoshka Quantization), with 240K training steps | | | | | | | | | | |
| 1-bit | 44.0 | 51.5 | 34.4 | 56.1 | 64.3 | 39.9 | 38.8 | 35.9 | 51.3 | 46.5 |
| 2-bit | 20.2 | 55.6 | 35.6 | 62.4 | 66.0 | 41.4 | 44.0 | 44.9 | 54.1 | 50.5 |
| 3-bit | 13.8 | 63.0 | 39.6 | 59.8 | 71.1 | 43.6 | 54.5 | 48.4 | 56.6 | 54.6 |
| 4-bit | 12.6 | 63.3 | 40.8 | 63.3 | 72.1 | 44.2 | 56.6 | 50.0 | 59.0 | 56.2 |
| Multi-level bit-width training (Ours), with 240K training steps | | | | | | | | | | |
| 1-bit | 20.7 | 54.5 | 34.7 | 46.7 | 65.4 | 41.1 | 41.9 | 39.3 | 55.2 | 47.4 |
| 2-bit | 13.3 | 63.1 | 39.8 | 61.8 | 71.3 | 44.3 | 55.4 | 50.6 | 57.1 | 55.4 |
| 3-bit | 11.9 | 66.2 | 42.4 | 63.6 | 72.5 | 44.3 | 58.9 | 51.6 | 57.7 | 57.1 |
| 4-bit | 11.6 | 66.6 | 43.5 | 63.6 | 72.8 | 44.3 | 59.4 | 51.0 | 58.2 | 57.4 |

*Table 12.* We report the complete comparison results of applying our approach with the Hadamard Transform, on top of the QuEST baseline. We evaluate LLaMA-3-1B with 1–2 bit weights and 4-bit activations. We evaluate the perplexity on WikiText-2 and average zero-shot accuracy across eight QA tasks. HT refers to the Hadamard Transform.

| LLaMA-3-1B | Wiki2 ($\downarrow$) | ARC-e | ARC-c | BoolQ | PIQA | SIQA | HellaSwag | OBQA | WinoGrande | Avg. Accuracy ($\uparrow$) |
|---|---|---|---|---|---|---|---|---|---|---|
| FP Model | 9.6 | 64.8 | 42.5 | 64.8 | 74.8 | 44.8 | 64.4 | 50.2 | 61.5 | 58.5 |
| **W1A4** | | | | | | | | | | |
| QuEST | 42.9 | 41.0 | 26.9 | 61.4 | 59.2 | 40.0 | 30.1 | 30.7 | 50.3 | 42.4 |
| WINQ w/ HT | **42.3** | 42.1 | 28.9 | 61.9 | 58.7 | 39.6 | 30.4 | 31.8 | 50.2 | **43.0** |
| **W2A4** | | | | | | | | | | |
| QuEST | 17.4 | 52.5 | 32.9 | 59.2 | 66.4 | 43.5 | 46.1 | 36.5 | 52.0 | 48.6 |
| WINQ w/ HT | **16.9** | 52.8 | 33.1 | 62.2 | 65.5 | 42.0 | 46.3 | 38.9 | 53.4 | **49.3** |

A recent work, Matryoshka Quantization (Nair et al., 2025), has explored this direction by training a single model across multiple bit-widths (2, 4, and 8 bits). Their method leverages the most significant bits of 8-bit integers to represent lower-bit integers. Conceptually, this approach can be seen as sharing clip values across different quantization levels. While their study focuses on min–max quantization, our exploration considers a more general setting that accommodates additional bit widths and alternative methods, such as LSQ.

Our preliminary study finds that noise injection stabilizes the training of multi-level bit-width QAT, yielding performance much closer to that of single-bit-level QAT while requiring only one round of training. Specifically, we train an LLaMA-1B model jointly at 1-, 2-, 3-, and 4-bit precision using the same computational budget as training a single-precision model. For each precision, we follow the quantization procedure in ParetoQ.

As shown in Table 11, the multi-bit model achieves performance comparable to single-precision models, with only a 1.7 increase in PPL and a 1.6% drop in test accuracy—while reducing training cost by 75%. For relative comparison, we evaluate against Matryoshka Quantization. At the same training cost, noise injection delivers better results, reducing PPL by 8.3 and increasing test accuracy by 2.4%.

*Table 13.* We report the full comparison of LLaMA-3-1B performance under 1- to 4-bit weight quantization. We evaluate the perplexity on WikiText-2 and average zero-shot accuracy across eight QA tasks.

| | Wiki2 (↓) | ARC-e | ARC-c | BoolQ | PIQA | SIQA | HellaSwag | OBQA | WinoGrande | Avg. Accuracy (↑) |
|---|---|---|---|---|---|---|---|---|---|---|
| FP Model | 9.6 | 64.8 | 42.5 | 64.8 | 74.8 | 44.8 | 64.4 | 50.2 | 61.5 | 58.5 |
| **W1A16** | | | | | | | | | | |
| RTN | 4.2e8 | 25.0 | 22.5 | 37.6 | 49.5 | 32.9 | 25.0 | 27.1 | 49.6 | 33.7 |
| GPTQ | 3.3e8 | 26.9 | 21.7 | 37.6 | 51.8 | 33.5 | 25.5 | 14.8 | 49.7 | 32.7 |
| SpinQuant | 2.4e8 | 25.0 | 22.5 | 37.6 | 49.5 | 32.9 | 25.0 | 27.1 | 49.6 | 33.7 |
| ParetoQ | 16.9 | 57.3 | 36.2 | 62.4 | 69.1 | 41.9 | 48.3 | 45.1 | 55.0 | 51.9 |
| WINQ | **15.3** | 59.7 | 37.0 | 61.3 | 69.5 | 42.6 | 49.8 | 46.1 | 54.4 | **52.6** |
| **W1.58A16** | | | | | | | | | | |
| RTN | 1.8e6 | 24.5 | 22.6 | 62.4 | 52.7 | 33.4 | 25.4 | 18.2 | 50.2 | 36.2 |
| GPTQ | 4.6e4 | 25.1 | 22.5 | 38.0 | 53.1 | 32.7 | 25.7 | 15.6 | 49.5 | 32.8 |
| SpinQuant | 2.2e3 | 25.0 | 21.5 | 37.6 | 51.9 | 33.4 | 25.3 | 17.2 | 49.1 | 32.6 |
| ParetoQ | 14.0 | 64.1 | 38.7 | 60.3 | 71.8 | 43.6 | 55.1 | 46.3 | 58.1 | 54.8 |
| WINQ | **12.9** | 65.0 | 39.7 | 62.1 | 72.9 | 44.3 | 56.2 | 47.1 | 57.4 | **55.6** |
| **W2A16** | | | | | | | | | | |
| RTN | 1.5e6 | 26.5 | 26.8 | 62.2 | 51.0 | 36.8 | 25.9 | 28.5 | 50.2 | 38.5 |
| GPTQ | 3.3e2 | 29.3 | 27.6 | 37.8 | 51.5 | 38.6 | 26.5 | 32.0 | 50.8 | 36.8 |
| AWQ | 2.0e5 | 27.4 | 26.0 | 48.9 | 50.2 | 37.0 | 25.7 | 24.4 | 51.5 | 36.4 |
| SpinQuant | 46.7 | 25.6 | 24.6 | 62.4 | 51.6 | 36.1 | 25.8 | 29.1 | 50.8 | 38.3 |
| ParetoQ | 12.5 | 64.8 | 41.7 | 62.8 | 73.1 | 44.0 | 56.6 | 52.0 | 58.5 | **56.7** |
| WINQ | **11.9** | 65.0 | 42.5 | 62.5 | 73.8 | 43.2 | 58.4 | 48.1 | 59.1 | 56.6 |
| **W3A16** | | | | | | | | | | |
| RTN | 30.9 | 28.9 | 25.0 | 55.9 | 53.5 | 37.8 | 30.1 | 28.9 | 50.6 | 38.8 |
| GPTQ | 68.6 | 37.4 | 27.3 | 43.1 | 58.4 | 39.2 | 37.1 | 32.4 | 53.8 | 41.1 |
| AWQ | 1.5e2 | 41.5 | 26.7 | 49.2 | 58.0 | 41.4 | 34.9 | 31.8 | 52.8 | 42.0 |
| SpinQuant | 12.6 | 56.9 | 34.9 | 61.0 | 69.3 | 42.0 | 53.4 | 41.2 | 56.2 | 51.9 |
| ParetoQ | **10.9** | 65.3 | 41.9 | 64.2 | 73.8 | 43.9 | 61.3 | 47.7 | 59.5 | 57.2 |
| WINQ | **10.9** | 65.9 | 43.2 | 63.9 | 74.4 | 44.8 | 61.2 | 49.0 | 60.2 | **57.8** |
| **W4A16** | | | | | | | | | | |
| RTN | 13.9 | 55.7 | 36.3 | 61.9 | 70.4 | 43.0 | 56.9 | 39.3 | 55.5 | 52.4 |
| GPTQ | 13.4 | 55.2 | 38.8 | 57.9 | 70.5 | 43.5 | 55.4 | 43.2 | 58.0 | 52.8 |
| AWQ | 12.2 | 63.4 | 40.0 | 63.5 | 73.4 | 44.5 | 60.5 | 45.8 | 60.3 | 56.4 |
| SpinQuant | 10.3 | 62.2 | 40.3 | 64.1 | 72.3 | 44.0 | 61.6 | 47.9 | 59.8 | 56.5 |
| ParetoQ | 10.3 | 67.4 | 43.4 | 64.4 | 74.8 | 44.4 | 63.5 | 50.4 | 61.4 | **58.7** |
| WINQ | **10.2** | 67.4 | 43.8 | 65.1 | 75.0 | 43.5 | 63.2 | 48.6 | 62.1 | 58.6 |

*Table 14.* We report the complete comparison results of LLaMA-3-1B and LLaMA-3-3B with 1–2 bit weights and 8-bit activations. We evaluate the perplexity on WikiText-2 and average zero-shot accuracy across eight QA tasks.

| LLaMA-3-1B | Wiki2 ($\downarrow$) | ARC-e | ARC-c | BoolQ | PIQA | SIQA | HellaSwag | OBQA | WinoGrande | Avg. Accuracy ($\uparrow$) |
|---|---|---|---|---|---|---|---|---|---|---|
| FP Model | 9.6 | 64.8 | 42.5 | 64.8 | 74.8 | 44.8 | 64.4 | 50.2 | 61.5 | 58.5 |
| **W1A8** | | | | | | | | | | |
| RTN | 4.7e8 | 25.0 | 22.5 | 37.6 | 49.5 | 32.9 | 25.0 | 27.1 | 49.6 | 33.7 |
| GPTQ | 3.8e8 | 26.9 | 21.7 | 37.6 | 51.8 | 33.5 | 25.5 | 14.8 | 49.7 | 32.7 |
| SpinQuant | 3.4e8 | 27.2 | 21.1 | 37.6 | 52.6 | 33.5 | 25.6 | 14.3 | 50.5 | 32.8 |
| ParetoQ | 23.3 | 55.4 | 31.4 | 61.2 | 66.5 | 40.5 | 38.2 | 39.8 | 52.5 | 48.2 |
| WINQ | **21.9** | 56.0 | 33.9 | 61.6 | 66.4 | 41.6 | 38.8 | 41.4 | 52.3 | **49.0** |
| **W1.58A8** | | | | | | | | | | |
| RTN | 1.8e6 | 24.5 | 22.3 | 62.4 | 52.7 | 33.4 | 25.4 | 18.4 | 50.2 | 36.2 |
| GPTQ | 7.5e4 | 24.8 | 22.2 | 38.0 | 52.2 | 32.5 | 25.4 | 17.0 | 49.4 | 32.7 |
| SpinQuant | 5.8e3 | 25.3 | 22.5 | 37.6 | 51.6 | 33.3 | 25.3 | 17.6 | 48.5 | 32.7 |
| ParetoQ | 18.2 | 59.6 | 36.4 | 60.8 | 69.8 | 43.2 | 47.6 | 43.6 | 54.3 | 51.9 |
| WINQ | **16.9** | 59.1 | 37.0 | 60.8 | 69.7 | 43.2 | 48.9 | 44.9 | 56.2 | **52.5** |
| **W2A8** | | | | | | | | | | |
| RTN | 1.5e6 | 24.5 | 23.1 | 62.4 | 52.3 | 33.6 | 25.4 | 17.6 | 50.3 | 36.1 |
| GPTQ | 3.8e4 | 26.9 | 21.7 | 37.6 | 51.8 | 33.5 | 25.5 | 14.8 | 49.7 | 32.7 |
| SpinQuant | 3.8e2 | 31.0 | 20.1 | 45.5 | 54.6 | 33.8 | 26.7 | 16.2 | 50.9 | 34.9 |
| ParetoQ | 16.9 | 59.5 | 37.1 | 61.9 | 69.5 | 42.5 | 48.1 | 44.5 | 54.5 | 52.2 |
| WINQ | **16.3** | 60.98 | 37.66 | 61.39 | 70.42 | 42.99 | 48.83 | 47.27 | 54.69 | **53.0** |
| LLaMA-3-3B | Wiki2 ($\downarrow$) | ARC-e | ARC-c | BoolQ | PIQA | SIQA | HellaSwag | OBQA | WinoGrande | Avg. Accuracy ($\uparrow$) |
| FP Model | 7.7 | 72.6 | 50.7 | 74.6 | 78.2 | 48.5 | 74.3 | 53.7 | 69.2 | 65.2 |
| **W1A8** | | | | | | | | | | |
| RTN | 7.3e7 | 25.0 | 22.5 | 37.6 | 49.5 | 32.9 | 25.0 | 27.1 | 49.6 | 33.7 |
| GPTQ | 5.9e7 | 26.6 | 21.9 | 37.6 | 52.5 | 33.7 | 25.6 | 15.0 | 49.4 | 32.8 |
| SpinQuant | 4.5e7 | 27.0 | 22.0 | 37.6 | 52.0 | 33.4 | 25.5 | 14.5 | 50.2 | 32.8 |
| ParetoQ | 15.7 | 63.1 | 39.1 | 63.0 | 70.9 | 42.6 | 50.6 | 46.7 | 56.5 | 54.1 |
| WINQ | **14.8** | 64.2 | 40.5 | 63.6 | 71.1 | 42.8 | 52.7 | 49.4 | 57.3 | **55.2** |
| **W1.58A8** | | | | | | | | | | |
| RTN | 7.9e5 | 24.7 | 23.4 | 37.6 | 52.9 | 33.8 | 25.5 | 17.4 | 50.3 | 33.2 |
| GPTQ | 2.7e5 | 25.3 | 22.9 | 37.6 | 53.7 | 34.4 | 25.3 | 15.6 | 49.5 | 33.1 |
| SpinQuant | 3.1e3 | 26.6 | 23.4 | 39.1 | 52.7 | 34.7 | 25.3 | 16.4 | 48.5 | 33.3 |
| ParetoQ | 13.1 | 67.9 | 42.0 | 53.9 | 72.0 | 43.9 | 58.2 | 49.6 | 60.2 | 56.0 |
| WINQ | **12.2** | 68.0 | 43.0 | 62.8 | 73.0 | 44.9 | 59.8 | 50.2 | 61.6 | **57.9** |

*Table 15.* We summarize the hyperparameters used for each setting. A batch size of $8 \times 8$ indicates 8 GPUs with a per-GPU batch size of 8. HT refers to the Hadamard Transform.

| Model | Weight | Activation | Learning rate | Batch size | Re-intialization interval $K$ | Interpolation scalar $\alpha$ | Standard deviation $\sigma$ |
|---|---|---|---|---|---|---|---|
| LLaMA-3-1B | 4-bit | 16-bit | $1 \times 10^{-5}$ | $8 \times 8$ | 40K | 0.2 | 0.0002 |
| LLaMA-3-1B | 3-bit | 16-bit | $1 \times 10^{-5}$ | $8 \times 8$ | 40K | 0.2 | 0.0002 |
| LLaMA-3-1B | 2-bit | 16-bit | $2 \times 10^{-5}$ | $8 \times 8$ | 80K | 0.2 | 0.001 |
| LLaMA-3-1B | 1.58-bit | 16-bit | $2 \times 10^{-5}$ | $8 \times 8$ | 80K | 0.2 | 0.001 |
| LLaMA-3-1B | 1-bit | 16-bit | $2 \times 10^{-5}$ | $8 \times 8$ | 60K | 0.4 | 0.001 |
| LLaMA-3-1B | 2-bit | 8-bit | $4 \times 10^{-5}$ | $8 \times 8$ | 60K | 0.2 | 0.001 |
| LLaMA-3-1B | 1.58-bit | 8-bit | $4 \times 10^{-5}$ | $8 \times 8$ | 80K | 0.2 | 0.001 |
| LLaMA-3-1B | 1-bit | 8-bit | $4 \times 10^{-5}$ | $8 \times 8$ | 60K | 0.4 | 0.001 |
| LLaMA-3-3B | 1.58-bit | 8-bit | $4 \times 10^{-5}$ | $8 \times 8$ | 60K | 0.2 | 0.001 |
| LLaMA-3-3B | 1-bit | 8-bit | $4 \times 10^{-5}$ | $8 \times 8$ | 60K | 0.2 | 0.001 |
| Qwen-3-1.7B | 2-bit | 8-bit | $2 \times 10^{-5}$ | $8 \times 8$ | 80K | 0.1 | 0.0002 |
| Qwen-3-1.7B | 1-bit | 8-bit | $2 \times 10^{-5}$ | $8 \times 8$ | 80K | 0.1 | 0.0002 |
| Qwen-3-0.6B | 2-bit | 8-bit | $2 \times 10^{-5}$ | $8 \times 8$ | 80K | 0.1 | 0.0002 |
| Qwen-3-0.6B | 1-bit | 8-bit | $2 \times 10^{-5}$ | $8 \times 8$ | 80K | 0.1 | 0.0002 |
| LLaMA-3-1B w/ HT | 2-bit | 4-bit | $4 \times 10^{-5}$ | $8 \times 8$ | 80K | 0.2 | 0.0002 |
| LLaMA-3-1B w/ HT | 1-bit | 4-bit | $4 \times 10^{-5}$ | $8 \times 8$ | 80K | 0.1 | 0.0002 |
| LLaMA-3-1B w/ HT | 2-bit | 4-bit | $4 \times 10^{-5}$ | $8 \times 8$ | 80K | 0.2 | 0.0002 |
| LLaMA-3-1B w/ HT | 1-bit | 4-bit | $4 \times 10^{-5}$ | $8 \times 8$ | 80K | 0.1 | 0.0002 |

*Table 16.* We report the complete comparison results of Qwen-3-1.7B and Qwen-3-0.6B with 1–2 bit weights and 8-bit activations. We evaluate the perplexity on WikiText-2 and average zero-shot accuracy across eight QA tasks.

| Qwen-3-1.7B | Wiki2 ($\downarrow$) | ARC-e | ARC-c | BoolQ | PIQA | SIQA | HellaSwag | OBQA | WinoGrande | Avg. Accuracy ($\uparrow$) |
|---|---|---|---|---|---|---|---|---|---|---|
| FP Model | 16.2 | 68.9 | 41.0 | 78.9 | 71.7 | 45.1 | 59.6 | 38.5 | 61.6 | 58.2 |
| **W1A8** | | | | | | | | | | |
| ParetoQ | 46.5 | 42.0 | 26.2 | 61.4 | 59.8 | 40.0 | 29.1 | 28.9 | 50.8 | 42.3 |
| WINQ | **45.9** | 42.4 | 26.4 | 61.4 | 59.6 | 40.3 | 29.0 | 30.1 | 49.7 | **42.4** |
| **W2A8** | | | | | | | | | | |
| ParetoQ | 22.2 | 52.9 | 32.6 | 62.4 | 65.1 | 42.7 | 41.2 | 32.8 | 53.1 | 47.8 |
| WINQ | **21.8** | 53.1 | 33.9 | 62.7 | 64.7 | 42.3 | 41.0 | 35.0 | 53.1 | **48.2** |

| Qwen-3-0.6B | Wiki2 ($\downarrow$) | ARC-e | ARC-c | BoolQ | PIQA | SIQA | HellaSwag | OBQA | WinoGrande | Avg. Accuracy ($\uparrow$) |
|---|---|---|---|---|---|---|---|---|---|---|
| FP Model | 53.6 | 35.3 | 65.6 | 67.4 | 42.6 | 46.5 | 36.1 | 56.6 | 50.5 | 20.1 |
| **W1A8** | | | | | | | | | | |
| ParetoQ | 64.0 | 37.3 | 24.2 | 61.5 | 56.6 | 39.5 | 27.3 | 33.0 | 50.6 | 41.2 |
| WINQ | **61.9** | 38.4 | 24.7 | 62.1 | 56.7 | 39.8 | 27.6 | 31.8 | 50.2 | **41.4** |
| **W2A8** | | | | | | | | | | |
| ParetoQ | **32.0** | 44.9 | 28.5 | 54.5 | 60.4 | 40.7 | 32.7 | 33.2 | 51.6 | 43.3 |
| WINQ | 32.1 | 44.7 | 28.3 | 60.2 | 60.0 | 40.7 | 32.5 | 33.4 | 51.1 | **43.9** |

