# OpenReview forum: "WinQ: Accelerating Quantization-Aware Training of Language Models Around Saddle Points"
_ICML.cc/2026/Conference — ICML 2026 regular_

### Official Review · Reviewer_rGhK · 2026-02-22

**Soundness:** 3
**Presentation:** 3
**Significance:** 3
**Originality:** 3
**Overall Recommendation:** 5
**Confidence:** 4

**Summary:**

This paper connects the current convergence difficulties of the QAT with a saddle point problem by illustrating that later training epochs have lower eigenvalue magnitudes. The proposed solution is applying noise injection with weight re-initialization, to improve convergence. Evaluation is completed in multiple extreme quantization settings across standard LLMs to illustrate that their method is competitive.

**Compliance With Llm Reviewing Policy:**

Affirmed.

**Final Justification:**

The authors have addressed my concerns, I maintain my score.

**Key Questions For Authors:**

In Nagel et. al [2022] (https://proceedings.mlr.press/v162/nagel22a/nagel22a.pdf) , the convergence issues of QAT are shown to be related to oscillations due to the STE. To what extent is this saddle point problem connected to artifacts related to the straight-through estimator?

It would be interesting to understand what types of noise injection and re-initialization, beyond the method the authors propose, alleviate the saddle point problem.

Could the authors comment on the applicability of this method to quantization aware distillation?

Child the authors as a zoomed in portion of Figure 2? It would be particularly interesting to understand how much the loss oscillates. Additionally, it would be very valuable to compare this to FP16 training.

How do we know that this observation is only for quantized training? The authors have not convinced me that this is a QAT-specific observation and couldn’t also be observed at half precision.

**Limitations:**

Yes

**Strengths And Weaknesses:**

The observation that eigenvalues become smaller during training is an important one, however, I wonder if the cause of this is actually an artifact of the STE and could be mitigated by choosing a different, more différentiable function. Nevertheless, this work has good observations and the illustration of eugénismes is intuitive.

Overall, the experimental results are extensive and wide, which gives credibility to the efficacy of this method. Adding additional baselines such as standard quantization aware training recipes from TensorRT would strengthen the evaluation, but are not strictly necessary. Furthermore, adding eugénisme visualizations for FL16 training as a side-by-side comparison would help strengthen the argument significantl.

The paper lacks a clear connection between the saddle point problem and the proposed solution. Couldn’t one just apply the methods from prior work, such as occasionally switching to FP16 training for 1 epoch?

---

> ### Author Rebuttal · Authors · 2026-03-31
>
> We thank the reviewer for the insightful comments! Below, we respond to each comment.
>
> **Evaluation of Hessian eigenvalues of non-STE-based methods.** (W1)
>
> Response: We further evaluate other gradient estimation methods, including the rotation trick (Fifty et al., 2025) and QuEST (Panferev et al., 2025). We train a 30M LLaMA-style model with 1–4 bit weight quantization. Across these methods, we observe consistent trends:
> - As training progresses, eigenvalues increasingly concentrate near zero, and their magnitudes decrease substantially (from around 180 at 1K steps to 3 at 10K steps in 1-bit).
> - Lower bit-widths exhibit smaller eigenvalue magnitudes (around 15 in 3-bit to 3 in 1-bit).
>
> These suggest that our observations are not limited to STE. We will include them in the updated paper.
>
> **Eigenvalue visualizations for FP16 training.** (W2 & Q5)
>
> Response: We include the Hessian spectrum for the FP16 model in Figure 6 (Appendix B). Compared to lower precisions, FP16 exhibits larger Hessian eigenvalue magnitudes (e.g., 15 in FP16 and 5 in 4-bit). This supports that the observed flatter loss landscape and slow convergence are severe in low-precision training.
>
> **Connection between the saddle point problem and our method.** (W3)
>
> Response: Our analysis shows that in low-precision training, training slows down as the model weights are stuck in regions with near-zero gradients and small Hessian eigenvalues. Our method directly targets this issue.
> - First, the weight interpolation reduces the distance between quantized and latent weights. We found that this moves the model weights to a region with larger Hessian eigenvalue magnitudes, thus accelerating the training. We can further show that **the weight interpolation is equivalent to a proximal update step of a regularized training objective, which provably increases the Hessian eigenvalues**.
> - Second, the noise injection is inspired by prior theoretical optimization works. We found that it leads to a large magnitude of eigenvalues and larger gradient norms, which accelerates the training.
>
> The supporting empirical results across 1-4 bits are shown in Section 3.2 and Appendix B.2. We will emphasize these points in the updated paper.
>
> **Evaluations of switching between FP16 and low-precision training.**  (W3)
>
> Response: Switching between FP16 and low precision can make training unstable, as the gradients differ substantially and disrupt optimizer statistics. In a preliminary experiment, we alternate between FP16 and 1-bit every 10K steps in training Llama-1B. We observe that switching to FP16 leads to much higher quantized model loss. While explored in CNNs, designing such methods for LLMs is an interesting future direction.
>
> **Discussing the weight oscillation problem (Nagel et al. 2022).** (Q1)
>
> Response: Thank you for suggesting this relevant work. This work identifies an oscillation phenomenon where weights repeatedly switch between adjacent quantization levels, leading to loss plateaus in training CNNs.
>
> Our work provides a different perspective on the convergence of quantized training by examining second-order gradients. One potential connection is that oscillations with a small change in loss suggest that Hessian eigenvalues are close to zero. This aligns with our observation of flatter loss regions in low bit-width settings. We will include a discussion of this work in the updated paper.
>
> **Discussion of other design choices.** (Q2)
>
> Response: For noise injection, alternatives include uniform noise scaled to quantization bins (similar to dithering) or injecting noise to gradients. For re-initialization, one could perturb weights along directions of negative eigenvalues or adapt interpolation per layer. These alternatives can require expensive computations (e.g., Hessian estimation) or additional hyperparameter tuning. In contrast, our method introduces negligible overhead and minimal tuning.
>
> **Discussion of quantization-aware distillation.** (Q3)
>
> Response: Our analysis naturally extends to this setting, where a quantized student is trained via KL divergence to a teacher. As the underlying quantization process remains largely the same, the student model is similarly prone to slow convergence at low bit-widths.
>
> **Evaluation of oscillation.** (Q4)
>
> Response: We analyze the fraction of weights within 1% relative distance to the grid boundaries, following Nagel et al. (2022). On LLaMA-1B, this fraction increases from 5% in 4-bit to 13% in 1-bit, indicating that more weights are prone to oscillation at lower bit-widths. This aligns with our observation of slower convergence in low-precision training. In contrast, FP16 exhibits a smooth distribution without such effects.
>
> **References:**
>
> Fifty et al. Restructuring Vector Quantization with the Rotation Trick. ICLR 2025
>
> Panferov et al. QuEST: Stable Training of LLMs with 1-bit Weights and Activations. ICML 2025
>
> Nagel et al. Overcoming Oscillations in Quantization-Aware Training. ICML 2022

---

> > ### Author Rebuttal · Reviewer_rGhK · 2026-04-03
> >
> > My questions have been fully resolved, the authors use QuEST and the rotation trick in their evaluation, which are important baselines beyond STE.

---

### Official Review · Reviewer_W3Dh · 2026-03-10

**Soundness:** 3
**Presentation:** 3
**Significance:** 4
**Originality:** 2
**Overall Recommendation:** 4
**Confidence:** 3

**Summary:**

The paper starts with a thorough study of gradient norm and Hessian eigenvalues throughout quantization-aware training with different bit-widths. The critical observation is that the Hessian eigenvalues cluster more and more around zero as the training progresses. This effectively means that the model reaches saddle points in the loss landscape.

To address this, the authors introduce WinQ, a two component method to increase the magnitude of Hessian eigenvalues during training. The first component is to replace parameters $W$ with $\alpha W + (1-\alpha) Q(W)$ once in a while. The second component is noise injection into the weights right before quantization. The authors show that this approach is also applicable to Hadamard-based quantization methods, and support their method with E2E results as well as ablations studies.

**Compliance With Llm Reviewing Policy:**

Affirmed.

**Final Justification:**

My concerns were almost fully  addressed during the rebuttal. I do think a more comprehensive comparison against the CAGE method is required. I am increasing my score to "Weak Accept".

**Key Questions For Authors:**

1. In Section 3.2, around line 253, you claim "Observe that this weight interpolation does not change the quantized weights and the corresponding gradients." Why is that? My understanding is that this statement is only true if you assume the grid/step size is fixed in the quantization function Q.
2. I generally struggle to understand why the interpolation increases the magnitude of Hessian eigenvalues. Assuming the statement above (in Question 1) is correct, that means right after the interpolation, the loss, gradient and the Hessian won't change at all. Do you mean this will improve Hessian's eigenvalues in the following steps?
3. Can the authors compare with CAGE [1]? Instead of the interpolation, CAGE pushes the weights towards the quantization grid through a regularization. I think this comparison is critical.
4. The quantization function Q can itself be viewed as noise injection. The amount of this noise becomes larger as the bit-width decreases. On top of this, you suggest introducing extra Gaussian noise. I would be interested to know the authors' insights on the interaction between these two sources of noise. For example, shouldn't this mean that less noise injection is required as the bit-width decreases?
5. Looking at the loss curves (e.g., Figure 4), there are two points during training where the loss drops significantly. Are those steps where the interpolation happens?
6. Regarding activation quantization, given that methods like QuEST go as low as W1A1, is there a specific reason why the authors don't go below 4-bit activations? Do you have any results for let's say W1A1?

Even though I think WinQ has the potential to be a good paper, I cannot suggest an acceptance in the current form. I would be open to increasing my score depending on how the rebuttal goes, especially regarding questions 1, 2, 3, and 6.

[1] https://arxiv.org/pdf/2510.18784

**Limitations:**

None from my side

**Strengths And Weaknesses:**

Strengths:
1. Systematic approach of a thorough study, detecting the problem, and tackling it directly.
2. Good presentation.
3. Ablation studies and sufficient experiments.
4. Near-zero overhead.

Weaknesses (see questions below for more details):
1. In my opinion, the use of interpolation and noise injection are under-justified.
2. Key related papers are missing.

---

> ### Author Rebuttal · Authors · 2026-03-31
>
> We thank the reviewer for the insightful questions! Below, we respond to each comment.
>
> **Clarifying the conditions under which the weight interpolation does not change the quantized weights and the gradients.** (Q1)
>
> Response: The reviewer is correct that this statement holds when the clipping grid or step size is fixed during the interpolation, such as in learnable step sizes. When the step sizes depend on the weights $W$, the interpolation can slightly change the quantized weights and the gradients. We will revise the text to clarify this condition.
>
> Importantly, our main findings of the Hessian spectrum do not rely on this assumption. Our intended point was to illustrate the intuition that the interpolation moves latent weights toward the quantized weights and reduces quantization error, without significantly increasing the loss.
>
> Empirically, even in settings where step sizes depend on $W$, we observe minor changes. Using the RMS-normalized quantization in QuEST, the norm of quantized weights changes by at most 3% after interpolation across 1–4 bits and interpolation scales.
>
> **Theoretical explanation for why the interpolation increases the magnitude of Hessian eigenvalues.** (Q2)
>
> Response: Recall that the weight interpolation updates as $W \leftarrow (1-\alpha) W + \alpha Q(W)$. While this can leave the quantized forward pass nearly unchanged, it moves the weights toward the quantization grid and alters the local curvature.
>
> **Theoretically, we can show that the weight interpolation is equivalent to a proximal update step in a regularized training objective**, which is the quantized training loss plus the distance from the latent weights to the quantization grid: $L_Q(W) + \frac{\gamma}{2} ||W - Q||^2$.
> - Under this, the Hessian becomes $\nabla^2 L_Q(W) + \frac{\gamma}{2} I$, as the $Q$ is considered locally constant during interpolation. This correspondingly increases the magnitude of Hessian eigenvalues.
> - Moreover, the interpolation coefficient is related to the regularization strength: $\alpha = \eta \gamma / (1 + \eta \gamma)$, where $\eta$ is the learning rate of the optimizer.
>
> This theoretical interpretation does not rely on any assumption of the quantization method. We will include this justification in the updated paper.
>
> **Comparison with the related work, CAGE [1].** (Q3)
>
> Response: Thanks for suggesting this insightful work. CAGE augments the STE gradient with a scaled quantization error. It is motivated by a multi-objective formulation that balances minimizing the training loss and the distance between latent and quantized weights.
>
> Our approach differs in both method design and analysis.
> - While CAGE modifies the backward update, our method applies weight interpolation and noise injection to regularize the loss Hessian.
> - We analyze the convergence of quantized training by examining the second-order gradients. In contrast, CAGE analyzes with the error feedback formulation.
> - Interestingly, our method is related to CAGE, via interpreting the interpolation as a proximal update toward the quantization grid. It also corresponds to regularizing the distance between latent and quantized weights.
>
> We also conducted a preliminary evaluation. Following CAGE setup, in training a 30M Llama-style model with the W4A4 setting and the QuEST base method, our method achieves a **1**% lower test perplexity. We observe similar gains in training the Llama-1B model. This may benefit from the noise injection technique. We will include this discussion in the revised paper.
>
> **Discussion of the random Gaussian noise and the quantization noise.** (Q4)
>
> Response: Both can be viewed as optimizing a noise-smoothed loss. However, the magnitude of quantization noise increases at lower bit-widths (its relative norm grows from 16% in 4-bit to 70% in 1-bit, as shown in Section 2). This makes the second-order approximation of the smoothed loss less accurate. In addition, quantization noise can concentrate in specific directions at low precision.
>
> In contrast, the second-order approximation remains accurate using a small Gaussian noise. Thus, the noise injection can be interpreted as regularizing the Hessian matrix.
>
> **In Figure 4, there are points where the loss drops significantly. Are they the interpolation steps?** (Q5)
>
> Response: Yes, these are the steps where we apply the interpolation.
>
> **Application to the setting of W1A1 using QuEST.**  (Q6)
>
> Response: Our method broadly applies across quantization settings. We further evaluate it with QuEST under the W1A1 setting. We train a 30M LLaMA-style model on the SlimPajama dataset. Our method achieves a **5**% reduction in perplexity and a **2**% reduction in test loss compared to QuEST. We will include these results in the revised paper.
>
> **References:**
>
> Tabesh et al. CAGE: Curvature-Aware Gradient Estimation For Accurate Quantization-Aware Training. 2025
>
> Panferov et al. QuEST: Stable Training of LLMs with 1-bit Weights and Activations. ICML 2025

---

> > ### Author Rebuttal · Reviewer_W3Dh · 2026-04-03
> >
> > I would like to thank the authors for their clarification. Regarding Q2, I understand that the loss and curvature on the new W point change; however, the forward/backward passes use Q(W) which is unchanged after interpolation, and since STE is used, this means the loss at the exact step in which the interpolation happens does not change at all. Anyhow, I think this will affect the trajectory in the subsequent steps, hence I don't have any major concerns there.
> >
> > In general, my concerns are fully resolved. I do think a more comprehensive comparison with CAGE is necessary though. All in all, I am increasing my score.

---

### Official Review · Reviewer_rc4s · 2026-03-12

**Soundness:** 3
**Presentation:** 3
**Significance:** 3
**Originality:** 3
**Overall Recommendation:** 4
**Confidence:** 4

**Summary:**

This paper investigates the slow convergence issue in quantization-aware training (QAT) for extremely low-bit large language models. The authors analyze the Hessian spectrum during QAT and show that model weights tend to converge to flat regions near saddle points, where many Hessian eigenvalues are close to zero, leading to slow optimization. Based on this observation, the paper proposes WINIQ, which periodically re-initializes weights through interpolation between full-precision and quantized weights, combined with noise injection to help escape saddle points. Experiments across multiple models, bit-widths, and quantization methods show that WINIQ can accelerate QAT by up to 4× and improve sub-4-bit quantization performance under the same training budget.

**Compliance With Llm Reviewing Policy:**

Affirmed.

**Final Justification:**

The authors’ response has addressed my concerns, and I tend to accept this paper.

**Key Questions For Authors:**

- The proposed method combines weight interpolation and noise injection. Could the authors provide more detailed ablation studies to clarify the individual contribution of each component to the final performance and convergence speed?

- The method is motivated by observations of the Hessian spectrum during QAT. Is there any theoretical guarantee or further analysis explaining why interpolating between full-precision and quantized weights leads to larger Hessian eigenvalues and faster convergence?

- The experiments are conducted on models up to a few billion parameters. How does the method scale to significantly larger models (e.g., 7B or 13B+) in terms of both effectiveness and training overhead?

- Although the paper reports fewer training steps, what is the actual wall-clock training speedup when accounting for the additional interpolation and noise operations?

**Limitations:**

yes

**Strengths And Weaknesses:**

Strengths

- The paper studies an important and practical problem, namely the slow convergence of quantization-aware training (QAT) for extremely low-bit LLMs, which is relevant for efficient model deployment.

- The paper provides empirical analysis based on Hessian spectrum observations to help explain why low-bit QAT may converge slowly.

- The proposed approach is relatively simple and easy to integrate into existing QAT pipelines without requiring major architectural modifications.

- Experiments are conducted on multiple models and bit-width settings, and the results consistently show improvements in training efficiency.

Weakness:

- The methodological novelty appears somewhat limited, as the core idea mainly combines weight interpolation (reinitialization) and noise injection, which are related to existing optimization techniques for escaping saddle points.

- The theoretical explanation remains relatively weak; while Hessian observations motivate the method, the paper does not provide a rigorous analysis of why the proposed interpolation consistently improves optimization.

- The experiments are limited to relatively small LLMs (e.g., up to a few billion parameters), and it is unclear whether the method scales effectively to larger models used in practice.

- The paper mainly focuses on training loss and convergence speed, while the impact on end-task evaluation and downstream capabilities could be explored more thoroughly.

---

> ### Author Rebuttal · Authors · 2026-03-31
>
> We thank the reviewer for the thoughtful feedback! We respond to each comment below.
>
> **Novelty of our method.** (W1)
>
> Response: Our work studies quantization-aware training from an optimization perspective. By analyzing the Hessian spectrum, we identify an underexplored phenomenon: the slow convergence of QAT arises from model weights converging to flat regions near saddle points. To our knowledge, this connection has not been explicitly documented in prior literature.
>
> Motivated by this, we propose weight interpolation. This is a novel technique for quantized training, and we found it is effective for accelerating quantized training. While noise injection has been studied in general optimization, our contribution is to demonstrate its effect in accelerating quantized training.
>
> **Theoretical explanation of why the interpolation improves optimization.** (W2 & Q2)
>
> Response: We found that the weight interpolation results in a larger magnitude of Hessian eigenvalues, consistently across 1 to 4-bit precisions as described in Section 3.2, which leads to accelerated training.
>
> Further, we can show that **the weight interpolation is provably equivalent to a proximal update step in a regularized training objective**, which is the quantized training loss plus the distance of the latent weights to the quantization grid: $L_Q(W) + \frac{\gamma}{2} ||W - Q||^2$. Under this objective, the Hessian becomes $\nabla^2 L_Q(W) + \frac{\gamma}{2} I$, as the $Q$ is the quantization grid in this step and is considered locally constant. This correspondingly increases the magnitude of Hessian eigenvalues. We will include the theoretical explanation in the updated paper.
>
> **The experiments on larger models.** (W3 & Q3)
>
> Response: Our method is not tied to a specific model size and can be directly applied to larger architectures. We included results on Llama-8B in Appendix B.4 under 2-bit weight quantization with 16-bit activations. Our method achieves a 1% relative improvement over the state-of-the-art baseline (ParetoQ, NeurIPS 2025). We will move these results to the main text and include additional discussion on scalability.
>
> **Performance of our method in downstream task evaluations.** (W4)
>
> Response: We clarify that our method not only improves convergence speed and training loss, **but also delivers significant improvement on downstream tasks**. As shown in Tables 1–3, we evaluate models using perplexity (PPL) on WikiText2 and average zero-shot accuracy across eight QA benchmarks. Under the same training budget, our approach consistently outperforms state-of-the-art quantized training methods, achieving up to **8.8**% relative improvement across 16 settings of bit-widths and quantization methods.
>
> **Ablation studies of the contribution of each component in our method.** (Q1)
>
> Response: We reported ablation results in Section 4.3. Specifically, we evaluate the contribution of each component (weight interpolation and noise injection) by removing them when training a LLaMA-1B model with 1-bit weights, respectively.
> - Compared to training without noise injection, our method improves performance by 4%.
> - Compared to training without weight interpolation, our method improves performance by 6.7%.
> - We also study hyperparameter sensitivity in Tables 7 and 8 (Appendix B), where we observe that smaller interpolation coefficients and larger re-initialization intervals generally yield better results.
>
> We will move more discussion of the ablation studies to the main text in the updated paper.
>
> **The wall-clock training speedup.** (Q4)
>
> Response: The additional computational cost of weight interpolation and noise injection is negligible compared to the base training. Therefore, the wall-clock speedup matches the reduction in training steps required to reach the same loss.
>
> We reported wall-clock runtime in Appendix B.5. On a setup with 26 CPUs and one A100 GPU, training LLaMA-1B with 1-bit weights takes 28.7 hours in total. The weight interpolation step incurs only 0.3 seconds of additional runtime, as it involves simple element-wise operations. Noise injection incurs an additional 0.2 GPU hours overall, which is less than 1% of total runtime.
>
> These results confirm that the overhead is minimal. We will emphasize this wall-clock analysis in the revised paper.
>
> **References:**
>
> Liu et al. ParetoQ: Scaling Laws in Extremely Low-Bit LLM Quantization. NeurIPS 2025.

---

> > ### Author Rebuttal · Reviewer_rc4s · 2026-04-07
> >
> > The authors’ response has addressed my concerns, and I tend to accept this paper.

---

### Official Review · Reviewer_Z1HS · 2026-03-12

**Soundness:** 3
**Presentation:** 3
**Significance:** 3
**Originality:** 3
**Overall Recommendation:** 5
**Confidence:** 3

**Summary:**

They analyze the QAT convergence by computing the Hessian spectrum of the model loss throughout quantization-aware training and find the key reason is that the model weights converge to flat surfaces near saddle points, with a large fraction of Hessian eigenvalues concentrated around zero, and the magnitude of both positive and negative eigenvalues decreases over training. Additionally, the convergence speed is slower in lower bit-widths with significantly smaller Hessian eigenvalue magnitude.

Thay propose an approach to accelerate quantized training with minimal overhead named WinQ. This approach periodically performs linear weight interpolation between the full-precision and quantized weights and computes gradients on noise-injected weights. Both techniques effectively regularize the Hessian and accelerate training, resulting in an algorithm broadly applicable to quantization methods.

**Compliance With Llm Reviewing Policy:**

Affirmed.

**Final Justification:**

My concerns have been well addressed. They contribute a good solution to the slow convergence of low-bit LLM QAT to saddle point problems via systematic Hessian spectrum analysis.

**Key Questions For Authors:**

1. A theoretical derivation of the optimal range of noise standard deviation (α≈0.4 and σ=0.001) for low-bit QAT could be discussed.
2.  Evaluation on inference-time performance on different hardware platforms (e.g., A100, V100 GPUs) is needed (inference latency, throughput, and memory footprint).
3. I encourage the author to compare more baselines and to discuss whether there are the same Hessian matrix problems for different training methods (such as the prediction matching ParetoQ and distillation methods like LLM-QAT)?

**Strengths And Weaknesses:**

Strengths
1. They make a foundational contribution by linking the slow convergence of low-bit LLM QAT to saddle point problems via systematic Hessian spectrum analysis.
2. WINQ is elegantly designed with minimal computational overhead.
3. The experiments cover activation/weight quantization combinations, and report both better convergence speedup and performance gains under the same training budget.

Weaknesses
The main weaknesses lie in theoretical proof, engineering implementation details, and ablation experiments. See Questions.

---

> ### Author Rebuttal · Authors · 2026-03-31
>
> We thank the reviewer for the thoughtful feedback! We respond to each comment below.
>
> **Theoretical justification for the range of hyperparameters.** (W1 & Q1)
>
> Response: Thank you for the insightful question. We discussed hyperparameter selection (including the interpolation scalar and noise standard deviation) in Section 4.3 and Appendix B.3. Here, we provide additional theoretical justification.
>
> First, the weight interpolation updates the weights as $W \leftarrow (1-\alpha)W + \alpha Q(W)$, which reduces the distance between the latent and quantized weights. **Theoretically, we can show that the weight interpolation is provably equivalent to a proximal update step on a regularized training objective**, which is the quantized training loss plus the distance of the latent weights to the quantization grid: $L_Q(W) + \frac{\gamma}{2} ||W - Q||^2$. Under this objective, the Hessian becomes $\nabla^2 L_Q(W) + \frac{\gamma}{2} I$, indicating an increase in Hessian eigenvalue magnitude. Correspondingly, the interpolation coefficient can be viewed as a control over the regularization strength: $\alpha = \eta \gamma / (1 + \eta \gamma)$, where $\eta$ is the learning rate of the optimizer. In practice, we found that an $\alpha$ between 0.2 and 0.6 often yields the best performance.
>
> Second, the noise injection is inspired by prior theoretical work, such as Jin et al. (2017). According to their theoretical framework, the optimal noise radius depends inversely on the dimensionality of the parameters and the Lipschitz constant of the Hessian. Intuitively, $\sigma$ should be large enough to help move along directions of negative curvature, and small enough to preserve the gradient information. While deriving an exact range in LLM training is challenging, we empirically found the best $\sigma$ often in the range from $5e^{-4}$ to $2e^{-3}$, which aligns with prior theoretical insights.
>
> We will include the theoretical justification in the updated paper.
>
> **Evaluation of inference-time performance on hardware platforms.** (W2 & Q2)
>
> Response: We note that our work focuses on accelerating convergence and improving performance during quantization-aware training. Since our method operates on top of existing quantization methods without modifying them, the resulting models have identical inference latency and memory footprint as the base quantization method.
>
> For completeness, we additionally benchmarked inference on LLaMA-1B using an NVIDIA H100 GPU (94GB memory), following the ParetoQ evaluation protocol. We use a single GPU with a context length of 2048, the vLLM Machete kernel for 4-bit, and a CUTLASS mixed-precision kernel for 2-bit inference. We observed that 2-bit quantization provides around 4$\times$ speedup over FP16 and a 1.2$\times$ speedup over 4-bit quantization. The memory usage scales proportionally to bit-width.
>
> While this demonstrates the inference benefits of the low-bit models, our primary focus is improving the optimization and convergence during quantized training.
>
> **Evaluation of Hessian spectrum under different training methods.** (W3 & Q3)
>
> Response: Thanks for the suggestion! Our main observations in this work are that as quantized training progresses, a large fraction of Hessian eigenvalues concentrates around zero, and the magnitude of both positive and negative eigenvalues decreases over training. Further, we found that the magnitude of Hessian eigenvalues is significantly smaller in lower bit precision, corresponding to the slower convergence.
>
> We further evaluated the Hessian spectrum using another state-of-the-art quantization method, QuEST (Panerov et al., ICML 2025). Following their setup, we train a 30M Llama-style model with 1-3 bit weight quantization. We observe consistent trends that Hessian eigenvalue magnitudes decrease during training and are substantially smaller at lower bit-widths. The Hessian eigenvalue magnitude decreases from 15 in 3-bit to 6 in 2-bit and 3 in 1-bit. We will include these results in the revised paper.
>
> For the prior distillation methods, it trains a quantized student model by minimizing the KL divergence between the teacher's output and the student's output. As the difference is the training loss and the underlying quantization methods remain largely the same, the student model is similarly prone to the same problem in low-precision training.
>
> **References**
>
> Jin et al. How to Escape Saddle Points Efficiently. ICML 2017
>
> Liu et al. ParetoQ: Scaling Laws in Extremely Low-Bit LLM Quantization. NeurIPS 2025
>
> Panferov et al. QuEST: Stable Training of LLMs with 1-bit Weights and Activations. ICML 2025

---

> > ### Author Rebuttal · Reviewer_Z1HS · 2026-04-03
> >
> > Thank you for your detailed response. Most of my concerns have been well addressed, including those related to the choice of the hyperparameters and applicability to more settings. For this reason, I am raising my score from 4 to 5.

---

### Decision · Program_Chairs · 2026-04-30

**Decision:**

Accept (regular)

**Comment:**

This paper presents a quantization-aware training (QAT) acceleration method, WinQ, for LLMs. The authors first empirically study gradient norm and Hessian eigenvalues during QAT process with different bit-widths, showing that the Hessian eigenvalues cluster more and more around zero as the training progresses. The authors claim that the model reaches saddle points in the loss landscape. To address this issue, WinQ periodically performs linear weight interpolation between the full-precision and quantized weights and computes gradients on noise-injected weights. The authors conducted experiments on multiple generation/common-sense reasoning QA tasks with different LLM architectures/scales to show the efficacy of WinQ.

The paper was initially/finally scored (4,4,3,4)/(5,4,4,5) by four reviewers, who mostly recognized the motivation, the technical soundness and the empirical results, and raised several concerns about 1) somewhat limited novelty; 2) the theoretical proof is weak;  3) existing methods closely related to this paper are not sufficiently discussed and compared; 4) missing practical deployment performance on different hardware platforms; 5) lacking experiments on large-sized LLMs; 6) narrow research scope, particularly convergence speed.

The authors provided detailed responses to address these concerns. All reviewers were satisfied with the rebuttal. Finally, two reviewer (Z1HS, rGhK) consistently gave the positive score Accept, and the other two reviewer (rc4s, W3Dh) consistently gave the weakly positive score Weak accept as the novelty/formulation of this paper are not strong enough and a more comprehensive evaluation is required. The AC read the paper, the reviews, the rebuttal and the reviewers' feedback. I agree with reviewers that this paper has a good motivation, and is technical sound and mostly shows good performance, and thus recommend an “Accept”. The authors are encouraged to include additional experiments, discussions and clarifications in the final version of paper.